# Structural basis of nucleosome assembly by the Abo1 AAA+ ATPase histone chaperone

Carol Cho[1,8]*, Juwon Jang[1,8], Yujin Kang[2], Hiroki Watanabe [3,4], Takayuki Uchihashi [4,5], Seung Joong Kim[6], Koichi Kato [3,4,7], Ja Yil Lee [2]* & Ji-Joon Song [1]*

The fundamental unit of chromatin, the nucleosome, is an intricate structure that requires histone chaperones for assembly. ATAD2 AAA+ ATPases are a family of histone chaperones that regulate nucleosome density and chromatin dynamics. Here, we demonstrate that the fission yeast ATAD2 homolog, Abo1, deposits histone H3–H4 onto DNA in an ATP-hydrolysis-dependent manner by in vitro reconstitution and single-tethered DNA curtain assays. We present cryo-EM structures of an ATAD2 family ATPase to atomic resolution in three different nucleotide states, revealing unique structural features required for histone loading on DNA, and directly visualize the transitions of Abo1 from an asymmetric spiral (ATP-state) to a symmetric ring (ADP- and apo-states) using high-speed atomic force microscopy (HS-AFM). Furthermore, we find that the acidic pore of ATP-Abo1 binds a peptide substrate which is suggestive of a histone tail. Based on these results, we propose a model whereby Abo1 facilitates H3–H4 loading by utilizing ATP.

[1] Department of Biological Sciences and KI for the BioCentury, Korea Advanced Institute of Science and Technology (KAIST), Daejeon, Korea. [2] School of Life Sciences, Ulsan National Institute of Science and Technology, Ulsan, Korea. [3] Institute for Molecular Science (IMS), National Institutes of Natural Sciences, Okazaki, Japan. [4] Exploratory Research Center on Life and Living Systems (ExCELLS), National Institutes of Natural Sciences, Okazaki, Japan. [5] Department of Physics, Nagoya University, Nagoya, Japan. [6] Department of Physics, Korea Advanced Institute of Science and Technology (KAIST), Daejeon, Korea. [7] Graduate School of Pharmaceutical Sciences, Nagoya City University, Nagoya, Japan. [8] These authors contributed equally: Carol Cho, Juwon Jang. *email: carol.cho@kaist.ac.kr; biojayil@unist.ac.kr; songj@kaist.ac.kr

Chromatin is a dynamic structure that undergoes significant structural changes during DNA replication, transcription, and repair. This necessitates the regulated assembly and disassembly of nucleosomes by correct deposition of histones or removal of histones from DNA. Nucleosome assembly occurs by a stepwise process where initial binding of H3–H4 histones onto DNA forms a tetrasome intermediate, and subsequently two H2A–H2B dimers are incorporated in a stepwise manner to form the hexasome and nucleosome, respectively. Nucleosome disassembly, on the other hand, is thought to proceed by the reverse pathway of assembly (reviewed in ref. [1]).

In order to carefully control nucleosome assembly and disassembly, histone chaperones act as molecular escorts that prevent histone aggregation and undesired nucleic acid interactions. Dysfunction of histone chaperones affects genome stability and gene expression, and can ultimately result in developmental disorders and cancer[2,3].

ATAD2 (also termed ANCCA) is a histone chaperone that has been implicated in nucleosome density regulation by histone H3–H4 loading or removal. It is highly overexpressed in various cancers and associated with poor patient prognoses[4–6]. Genetic evidence relying on ChIP-seq and RNA-seq experiments suggests that ATAD2 increases chromatin dynamics and gene transcription[7]. The budding yeast homolog of ATAD2, Yta7, localizes to highly transcribed regions and decreases nucleosome density, implying that Yta7 evicts H3–H4 to facilitate transcription[8]. In contrast, it was reported that the fission yeast homolog, Abo1, promotes nucleosome occupancy and positioning, potentially by catalyzing assembly of H3–H4 onto DNA[9]. Thus, the precise biological role of the ATAD2 family is still debatable and remains to be determined.

Although histone chaperones responsible for nucleosome assembly and disassembly usually do not contain ATPase domains, ATAD2 is unique because it is the only known AAA+ ATPase histone chaperone. AAA+ ATPases are a highly conserved family of oligomeric ring-shaped motors that utilize ATP energy to remodel substrates[10,11]. The most classical functions associated with AAA+ ATPases are substrate unfolding or disruption of protein–protein interactions—activities that are performed by pulling on substrates through the central pore of the AAA+ ring. ATAD2 has two AAA+ domains in addition to a bromodomain that recognizes histones[12], thus spawning the idea that ATAD2 might perform work on histones analogous to other AAA+ ATPases. Despite such speculation, the biochemical activity and overall structure of any ATAD2 family member have been elusive.

Here, we demonstrate ATP-dependent histone H3–H4 deposition onto DNA by Abo1, the fission yeast ortholog of ATAD2, using an in vitro single-molecule imaging technique, the DNA curtain. We solve the cryo-EM structures of Abo1 in three different nucleotide states (apo, ADP, and ATP), revealing a major reorganization of the AAA+ domains from an open spiral to a closed ring—a structural change that can also be recapitulated in real time by high-speed atomic force microscopy (HS-AFM). Furthermore, we discover a mechanism by which Abo1 accommodates histone substrates, ultimately allowing it to function as a unique energy-dependent histone chaperone.

## Results

**Abo1 is an ATPase that interacts with histone H3–H4.** In order to directly determine the activity of an ATAD2 family ATPase in vitro, we recombinantly purified a near full-length version of *Schizosaccharomyces pombe* Abo1 where only the acidic N-terminus was truncated. The construct encompassed the two AAA+ domains, the bromodomain, and a C-terminal domain, all of which are structurally well conserved in the ATAD2 family (Fig. 1a and Supplementary Fig. 1). The profile of the purified

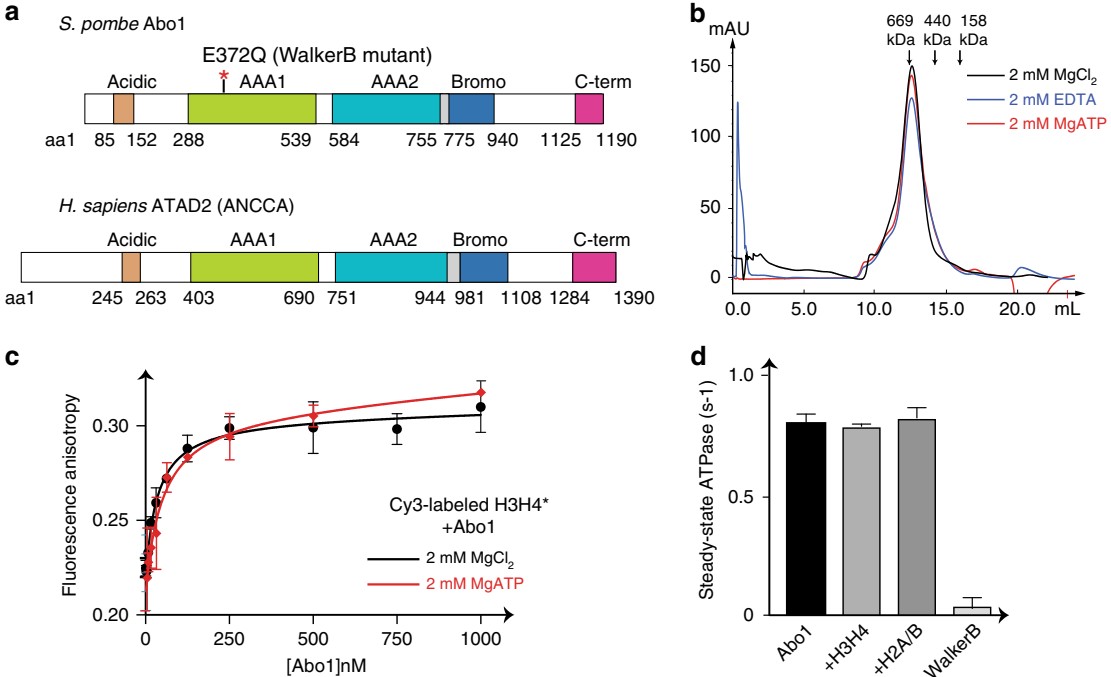

**Fig. 1 Recombinant Abo1 is an ATPase that binds histone H3–H4. a** Conserved domain organization of S. *pombe* Abo1 and human ATAD2. We term the region between the AAA2 domain and bromodomain (shown in gray) the "linker arm" based on structural data shown below (Fig. 3b, c). **b** Gel filtration profile of Abo1 over a Superose6 column under different buffer (2 mM EDTA, 2 mM MgCl₂, and 2 mM MgATP) conditions. **c** Binding of Abo1 to Cy3-labeled H3-H4 measured by fluorescence anisotropy assays. The Kd of Abo1 is 23 ± 13 nM. Error bars represent the standard error of the mean (SEM) for three experiments with different preparations of protein. **d** Steady-state ATPase rate of Abo1 in the presence of histone substrates. Error bars represent SEM for three experiments with different preparations of protein.

recombinant protein exhibited a homogeneous distribution on a gel filtration column which corresponded to the size of a hexamer, regardless of the presence or absence of nucleotide (Fig. 1b). This was in contrast to other AAA+ ATPases that usually form monomers in the absence of nucleotide.

Consistent with previous genetic studies[8,9], we found that recombinant Abo1 bound specifically to histone H3–H4 with nanomolar affinity (Kd of ~ 23 ± 13 nM, Fig. 1c) using fluorescence polarization with Cy3-labeled histone H3–H4 (Cy3-H3–H4*). The affinity did not change significantly in the presence of nucleotide, suggesting that the Abo1-histone interaction is not particularly sensitive to ATP. Enzymatically, Abo1 displayed a steady state ATPase rate of 0.83 ± 0.07 ATP/hexamer/s) that was unchanged by the addition of histones (Fig. 1d). These data

together show that Abo1 is an ATPase that tightly interacts with histone H3–H4.

**ATP hydrolysis-dependent H3–H4 loading onto DNA by Abo1.** Although we confirmed that Abo1 is an ATPase interacting with H3–H4, it was unclear whether Abo1 is involved in assembly or disassembly of histones. To directly visualize the process of histone H3–H4 loading or unloading on DNA, we adopted a single-tethered DNA curtain assay, which allows real-time imaging of fluorescently-labeled proteins bound to individual DNA molecules in a microfluidic chamber using total internal reflection fluorescence microscopy (Fig. 2a)[13,14]. By adding Cy5-labeled H3–H4 to DNA curtains and switching flow

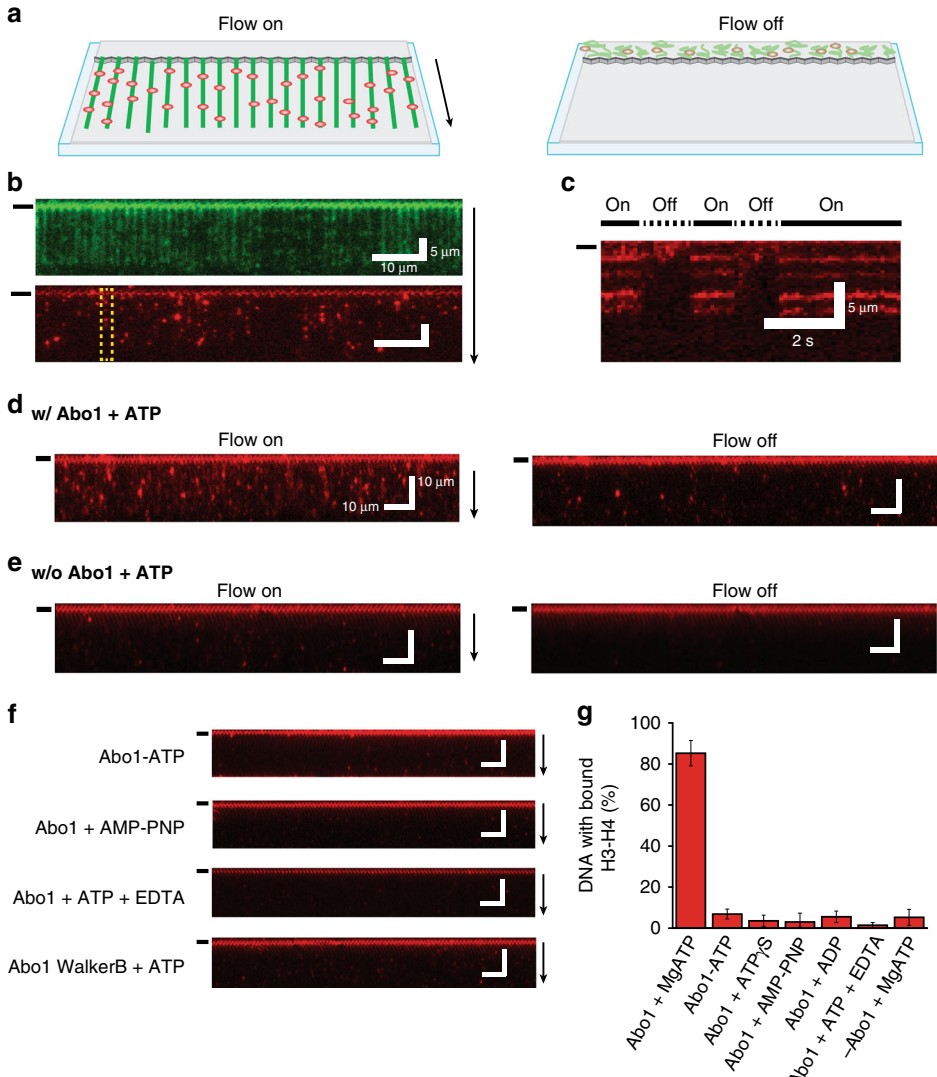

**Fig. 2 A single-molecule DNA curtain assay shows ATP hydrolysis-dependent H3–H4 loading onto DNA by Abo1. a** Schematic diagram of the single-tethered DNA curtain assay for histone deposition. In the presence of flow, DNA molecules are aligned and extended at the barrier, and can be visualized by TIRF microscopy. When the flow is stopped, DNA molecules recoil out of the evanescent field. **b** (Top) An image of DNA molecules (green) stained with YOYO-1. (Bottom) An image of Cy5-labeled H3–H4 (red) loaded by Abo1 in the presence of ATP. **c** Kymograph of a single DNA molecule in the yellow dashed box in **b**. When the flow is turned off, the Cy5 fluorescence signals disappear, ensuring that H3–H4 is specifically bound to DNA and not the surface. **d**, **e** Cy5-labeled H3–H4 loading onto DNA observed by the DNA curtain assay in the presence (w/ Abo1 + ATP) (**d**) or absence (**e**) of Abo1 (w/o Abo1 + ATP). In all images above, black bars left to image and arrows right to image indicate barrier position and flow direction, respectively. Scale bars represent 10 μm length unless indicated. **f** The ATP hydrolysis-dependence of Abo1 histone deposition activity as shown by DNA curtain assays with flow. **g** Quantification of histone H3–H4 deposition activity under different nucleotide conditions by measuring the fraction of DNA with bound H3–H4. Error bars represent the standard deviation (SD) of three experiments, and 100–200 molecules per experiment were analyzed for the quantifications.

on and off, we were able to image the specific attachment of histones to DNA (Fig. 2b, c). When histone H3–H4 mixed with Abo1 (w/Abo1) and ATP (+ATP) was flowed into a DNA curtain, DNA molecules were decorated with histones as shown by the appearance of Cy5 signal (red), whereas the absence of Abo1 (w/o Abo1) showed nearly no DNA binding (Fig. 2d, e and Supplementary Movies 1–2).

Because Abo1 is an AAA+ ATPase, we next asked whether the H3–H4 loading activity of Abo1 is nucleotide-dependent. While we observed that ~85% of DNA molecules were loaded with H3–H4 in the presence of Abo1 and ATP, less than 10% of DNA molecules were bound to H3–H4 in the absence of ATP, suggesting that ATP is required for Abo1-dependent H3–H4 loading onto DNA (Fig. 2f, g). The addition of ADP, non-hydrolyzable ATP analogs (ATPγS and AMPPNP), or the use of an Abo1 Walker B mutant (E372Q) also inhibited H3–H4 loading onto DNA, indicating that ATP hydrolysis, rather than ATP binding, is critical for Abo1 to deposit histones onto DNA. These data show that Abo1 directly loads histone H3–H4 onto DNA in the presence of ATP.

We then analyzed the positions of loaded H3–H4 by Abo1 on lambda DNA (λ-DNA) to examine if Abo1 has a preference for specific DNA sequences. The H3–H4 binding distributions showed no preference for binding position, demonstrating that Abo1 deposits H3–H4 on DNA in a sequence-independent manner (Supplementary Fig. 2a) as expected for a general histone chaperone.

In order to probe the characteristics of the histone–DNA complexes assembled by Abo1, we next performed MNase digestion assays (Supplementary Fig. 2b), to see if histones loaded by Abo1 could confer MNase protection. CAF-1, a well-characterized histone chaperone that is known to assemble tetrasomes by deposition of H3–H4 onto DNA, conferred protection of 70–80 bp DNA fragments, as expected for a histone chaperone that promotes tetrasome assembly[15].

Abo1, in contrast, did not protect 70–80 bp DNA fragments but showed a distinct MNase digestion pattern, where long DNA fragments (~150 bp) and short DNA fragments (<50 bp) were protected. We also observed protection of the short DNA fragments with Abo1 alone, implying that Abo1 binds to DNA even in the absence of histones, and that the long protected fragments result from DNA binding of Abo1–H3–H4 complexes. Taken together, these results support the idea that Abo1 loads histones onto DNA in a distinct manner from conventional tetrasomes.

Besides loading histones onto DNA, we also asked whether Abo1 can unload histones from DNA curtains, because studies of the human and budding yeast counterparts of Abo1 show that nucleosome density decreases in the absence of these proteins. We assembled H3–H4 onto DNA with yeast CAF-1 (yCAF-1) and examined if the level of assembled H3–H4 decreased when Abo1 was added. When Abo1 is added to DNA-histone curtains, we found no significant reduction in the fluorescence intensity of histones nor the fraction of histone-bound DNA (Supplementary Fig. 3 and Supplementary Movies 3–4). Thus, Abo1 catalyzes histone H3–H4 deposition onto DNA but does not directly promote removal of H3–H4 from DNA, which is consistent with previous in vivo studies showing that Abo1 is involved in chromatin assembly and organization[9].

**Cryo-EM structure of Abo1 in the ATP state**. To gain further insight into the molecular mechanism by which Abo1 couples ATP hydrolysis to histone loading, we determined the cryo-EM structures of Abo1 in three different nucleotide states (ATP-bound and ADP-bound states, and the apo state). We first solved

the cryo-EM structure of Abo1 in an ATP-bound state to 3.5 Å resolution using an ATP-hydrolysis deficient Abo1 Walker B mutant (E372Q) (Fig. 3, Supplementary Fig. 4 and Supplementary Table 1). The high quality of the cryo-EM map enabled us to resolve most amino acid side chains (Supplementary Fig. 5), and build a de novo model of Abo1 without applying knowledge from previously solved AAA+ structures.

The atomic resolution model revealed two layers of AAA+ domains arranged in a hexameric spiral (Fig. 3a), with the AAA1 domains forming the top layer and the AAA2 domains the bottom layer. The C-terminal domain was positioned underneath the AAA2 domain. We assigned the region of weak density lying above the plane of AAA1 to the bromodomain, because this is the only globular domain unaccounted for in our structural model.

Densities for six bound nucleotides were found at the interface of AAA1 protomers (Supplementary Fig. 6). We assigned five of the nucleotide densities as ATP, but the identity of one nucleotide (in subunit A) could not be determined with certainty although we modeled this nucleotide as ADP based on superposition with the ADP state (see below). No nucleotide density was found at the interface of the AAA2 domains, consistent with the fact that AAA2 has no consensus Walker A or B motifs and is presumably catalytically inactive.

The structural fold of the Abo1 AAA+ domain is overall similar to those of other AAA+ proteins in the classical clade[10], where the nucleotide binding α/β domain (NBD) and the helix bundle domain (HBD) are well conserved. However, superposing Abo1 with other AAA+ ATPases highlights non-canonical features that distinguish it from other AAA+ ATPases (Fig. 3c).

First, Abo1 subunits have a molecular clasp where the "knob" of one subunit locks into a "hole" of the neighboring subunit by an electrostatic interaction (Fig. 3b–d, Supplementary Fig. 7). The "knob" is formed by a helix-turn-helix insert between the first helix (α0) and first sheet (β1) of AAA2 NBD (Supplementary Fig. 8), a feature that is not observed in other AAA+ ATPases. The "hole" is formed by the AAA1 HBD, AAA2 HBD, and a linker that proceeds the AAA2 HBD (Fig. 3c). This unique addition, which we name the "linker arm" (Figs. 1a and 3b, c), stabilizes the hexameric conformation and explains how Abo1, in contrast to other AAA+ ATPases, can exist as a stable hexamer even in the absence of nucleotide (Fig. 1b).

Second, the AAA2 HBD consists of two distinct parts that originate from distant parts of the primary structure. Helix 5 (α5) of AAA2 HBD extends directly out of the AAA2 NBD, but helices 6–8 (α6–α8) of the AAA2 HBD originate from the Abo1 C-terminal domain (Fig. 3b, e). Thus, what was originally identified as the C-terminal domain[12] is in fact a part of AAA2 HBD.

Finally, Abo1 has two linker regions of significant length that join the AAA+ domains and substrate binding bromodomains in an unusual arrangement (Fig. 3b). A ~40aa linker arm (aa 755–775) protrudes out from the first helix of AAA2 HBD on the bottom of the AAA+ ring and extends toward the substrate binding bromodomain on top of the AAA+ ring. Another linker that we observe weak density for, presumably extends out of the bromodomains on the top of the molecule, and inserts back into the second helix of AAA2 HBD on the bottom of the AAA+ ring. These two linkers form structures that span the side of the AAA+ ring in opposite directions, and connect the bromodomains and AAA2 domains.

**Nucleotide-dependent structural changes of Abo1**. To dissect nucleotide-dependent structural changes of Abo1, we next determined the cryo-EM structure of Abo1 in the ADP state to 4.4 Å resolution. In addition, we determined two cryo-EM

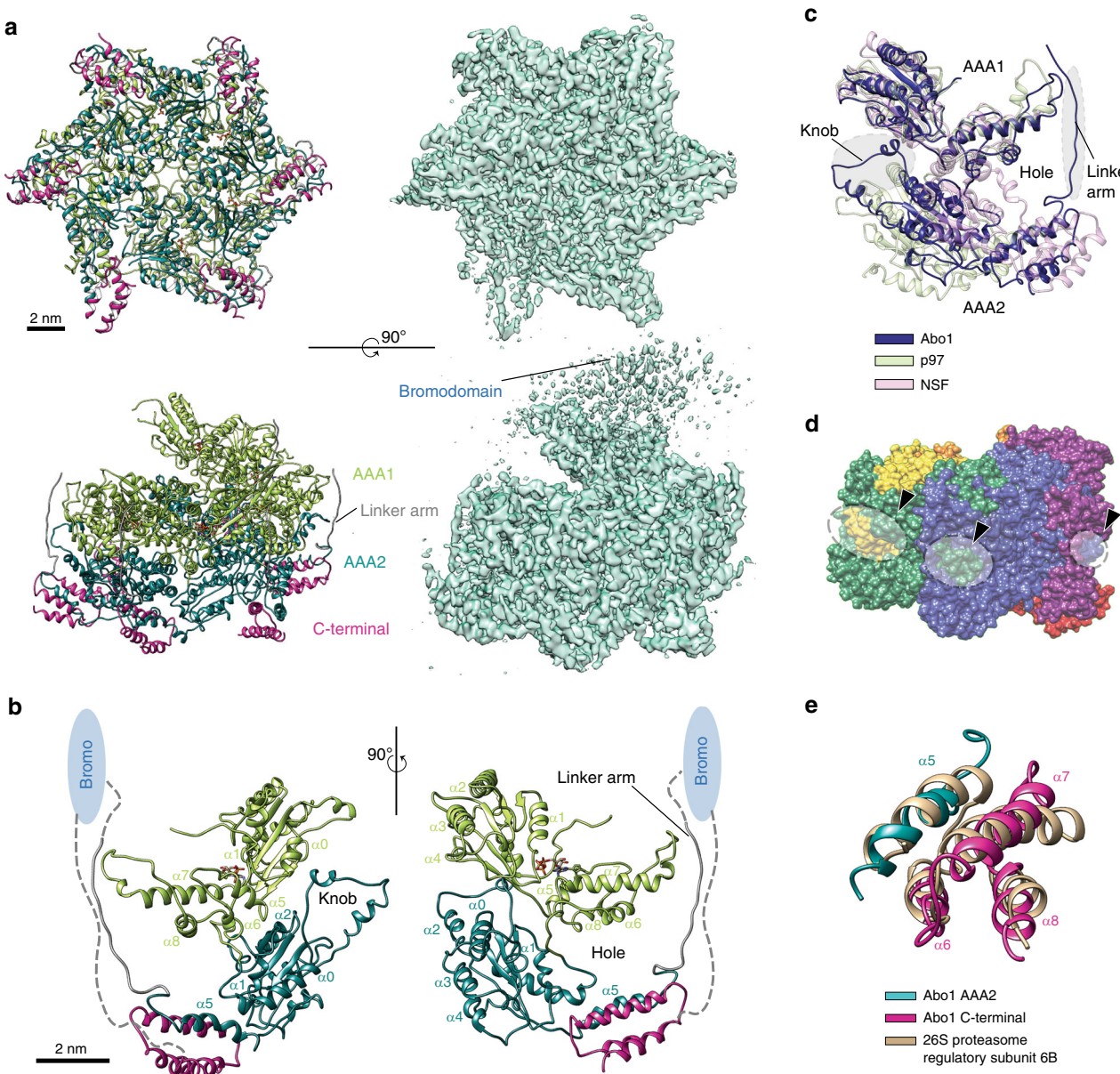

**Fig. 3 Cryo-EM structure of Abo1-Walker B mutant in the ATP state shows unique AAA+ structural organization. a** An atomic model (left) and cryo-EM map (right) of ATP-Abo1 Walker B mutant viewed from "bottom" (AAA2 side, top panel) and "side" of the AAA+ ring (bottom panel). The AAA1 domain is colored in light green, the AAA2 domain in teal, and the C-terminal domain in magenta. From the side view, the bromodomain, AAA1, AAA2, and the C-terminal form four tiers. The scale bars indicate 2 nm (20 Å). **b** The structure of a single Abo1 monomer within the Abo1 hexamer. Helices of AAA1 and AAA2 NBD and HBD are labeled according to AAA+ structural convention. Approximate bromodomain position based on electron density maps is depicted as a cartoon oval. The connectivity of the AAA+ domain, bromodomain, and C-terminal domain based on the structure and cryo-EM map is indicated as dotted lines. **c** Superposition of the Cα's of the Abo1 backbone (dark blue) with the AAA+ ATPases p97 (light green, PDB ID: 5FTM) and NSF (pink, PDB ID: 3J94), aligned with respect to AAA1. The unique inserts of Abo1—the AAA2 knob and the linker arm proceeding AAA2 α5—are highlighted by shaded ovals. The AAA2 domains do not align well due to the variable angle between AAA1 and AAA2. **d** The "knob and hole" packing of Abo1 subunits where the knob of one subunit inserts into the hole of the adjacent subunit (black arrowheads and dotted circles). Abo1 subunits are colored by chain. **e** The "split" AAA2 helical bundle domain (HBD) of Abo1, where α5 of AAA2 (teal, aa 734–750) interacts with three helices (α6–α8) of the C-terminal domain (magenta, aa1129–1185). AAA2 HBD is superposed onto its closest structural relative, 26 S proteasome AAA-ATPase subunit 6B (tan, PDB ID: 5ln3), showing that the three-dimensional structure of Abo1 HBD is conserved with other AAA+ HBD's despite unique helix connectivity.

structures in apo-states. One structure was determined to 4.3 Å resolution, which is similar in resolution to ADP-Abo1. The other structure that showed clear features of the bromodomain was determined to 6.9 Å resolution from data collected using a Volta phase plate. Therefore, we used the 4.3 Å resolution structure of apo-Abo1 for comparing with ATP- or ADP- bound states, and referred to the lower resolution structure of apo-Abo1 for

describing the bromodomain (Fig. 4a, Supplementary Figs. 9, 10 and 11 and Supplementary Table 1). We then performed flexible fitting of the ATP-model into the ADP- and apo- cryo-EM maps with MDFF[16] (Fig. 4b and Supplementary Fig. 13). Comparison of the ATP-Abo1 Walker B mutant structure with the ADP- and apo-states revealed striking differences in the subunit arrangement, heights and individual subunit conformation (Fig. 4b),

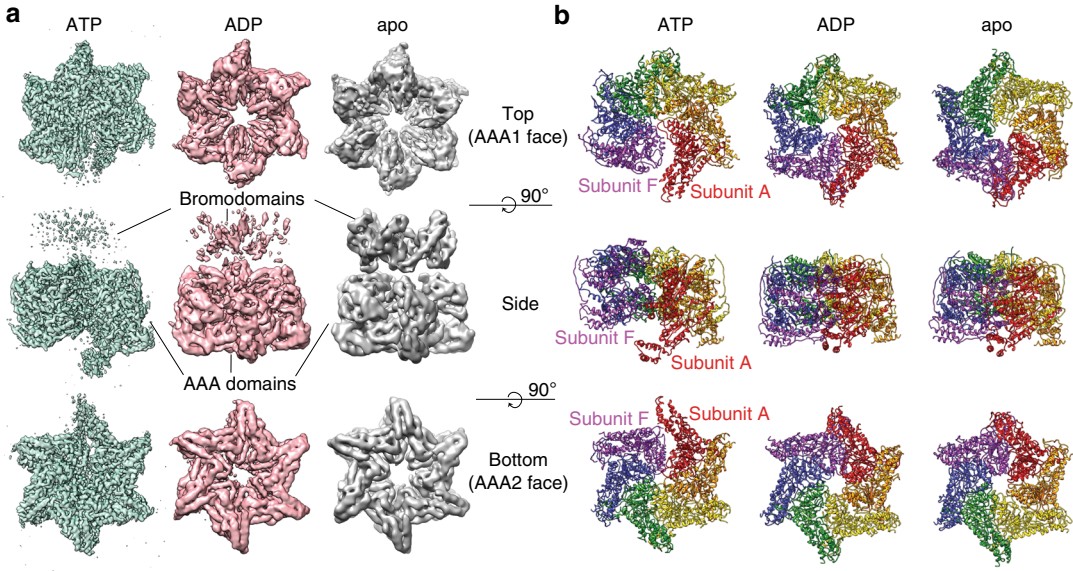

**Fig. 4 Comparison of Abo1 cryo-EM structures in the ATP and ADP states reveal a hexameric spiral-to-ring transition. a** Cryo-EM maps of Abo1 in the ATP, ADP, and apo states showing the "top" (AAA1), "side", and "bottom" (AAA2) face of the AAA+ hexamer. Resolution of ATP, ADP, and apo structures are at 3.5 Å, 4.4 Å, and 6.9 Å respectively. Electron density above the AAA+ ring is assigned as the bromodomain. The top view is clipped underneath the bromodomain to show the upper surface of the AAA1 domains. A top view of the apo-Abo1 bromodomain is shown in Supplementary Fig. 12. **b** Structures of Abo1 in the ATP state, and flexible-fitted Abo1 models in the ADP and apo state. The Abo1 hexamer is colored according to subunit (chains A–F).

while the structures of the ADP- and apo-states were similar to each other.

In the ADP state, the subunits assumed similar heights forming a symmetric planar ring, whereas in the ATP-state, the subunits were staggered in height and shifted towards the center forming an asymmetric spiral with a smaller pore. The most prominent structural change could be observed at the interface of subunits A and F where in the ADP state (Fig. 5a, b), these two subunits maintained similar heights with close inter-subunit packing, while in the ATP state, the subunits were staggered in height and separated by ~40 Å.

To further dissect the details of this conformational change we aligned the individual subunits of Abo1 in the ADP- and ATP-states. Individual subunits in the ADP state did not show any significant differences in their structures, supporting the idea that ADP-Abo1 is largely symmetric. However, individual subunits in the ATP state displayed a marked difference where the AAA1 domain and the linker arm progressively shift away from the AAA2 domain as the subunits rise in height along the spiral axis (Fig. 5c). When comparing the bottom-most subunit of the spiral (subunit A) with the top-most subunit (subunit F), the distance between the AAA1 and AAA2 NBD increases by 4.9 angstroms, and the axis between the AAA1 and AAA2 NBD tilts 10 degrees with respect to the AAA2 HBD. This implies that the junction between AAA1 and AAA2 is flexible and contributes to the variability of AAA1-AAA2 angles. In addition, linker arm flexibility can change the shape of the "hole" in the interlocking "knob-and-hole" structures. Notably, the density for the linker arm in the bottom-most subunit (subunit A) and the density for the "knob" in the top-most subunit (subunit F) were not discernible, as the subunits do not interlock at this interface.

When comparing the two extreme conformations of subunits (subunit A and subunit F) in the ATP-bound structure with the conformation of the subunit in the ADP-bound structure, we found that the bottom-most subunit in the ATP-bound structure (subunit A) superimposed well with the subunit in the ADP-bound structure (Fig. 5d), while the top-most subunit (subunit F)

diverged significantly from the ADP-conformation. All other subunits assumed positions that clustered in between these two extremes. Therefore, the variability observed in the individual ATP subunits also provides the basis for the conformational change between the ATP and ADP states. Based on this information, the bottom-most subunit is likely in a post-hydrolysis ADP state, which is also in agreement with other asymmetric spiral AAA+ ATPases[17].

Lastly, but more interestingly, in contrast to the ATP-Abo1 structure, the bromodomain density was stronger and more symmetric in the ADP- and apo- structures (Fig. 4a). Especially in the low resolution cryo-EM structure of the apo-state, six lobes of density that match the expected dimensions of a bromodomain were arranged in a hexameric ring around a central lobe of density. On the outer face of the bromodomain ring, characteristic helical densities that run diagonal to the central ring axis could be observed (Supplementary Fig. 12). However, due to the low resolution of this region, we were unable to fit bromodomain homology models with confidence, nor assign secondary structures. In addition, we also observed an extra density at the center of the bromodomains, although it is unclear whether this is part of Abo1 or a bound substrate. Collectively, the cryo-EM structures in three different nucleotide states reveal substantial structural change in Abo1 upon ATP hydrolysis which likely plays a critical role in loading histones on DNA, and also suggest that structural changes in the AAA+ ring may be coupled to changes in the organization of the bromodomains.

**ATP-dependent structural change of Abo1 observed by HS-AFM.** In order to understand the conformational dynamics of Abo1 in real time, we proceeded to observe Abo1 by HS-AFM. Upon adsorption of Abo1 onto a chemically modified mica surface, Abo1 appeared as hexameric rings (Fig. 6a). The rings were subjected to analysis of the full width at half maximum (FWHM), showing a single Gaussian distribution with a mean ± SD of $19.8 ± 2.6$ nm (Fig. 6b). The FWHM and height of the rings

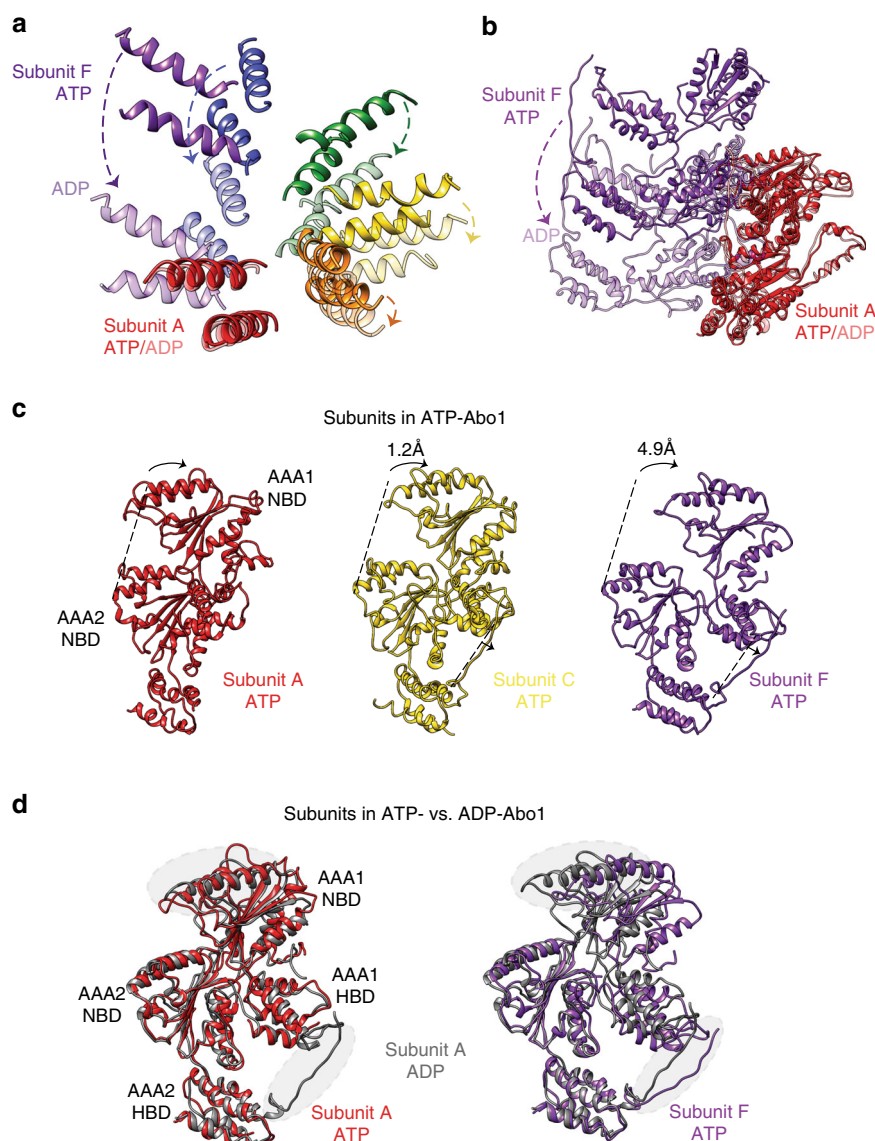

**Fig. 5 The hexameric spiral-to-ring transition of Abo1 is mediated by subunit and AAA1/2 domain movements. a** Rearrangement of Abo1 hexamer subunits from a spiral staircase (ATP state, bold colors) to a planar ring (ADP state, faded colors), as represented by the change in positions of AAA1 NBD α2 and α3. The two structures are superimposed by alignment at subunit A, and arrows depict the degree of movement of each subunit from the ATP to ADP state. **b** ATP (bold colors)-to-ADP (faded colors) structural transition of Abo1 subunit A and F. Structures are aligned by the AAA2 domain of subunit A. **c** Comparison of individual subunits of ATP-Abo1 (subunits A, C, and F) juxtaposed in the same configuration when aligned by AAA2. AAA1 NBD and the linker arm move away from AAA2 NBD. AAA1 NBD and AAA2 NBD centroid positions are further apart in subunits C and F by 1.2 and 4.9 Å, respectively, when compared to subunit A. The angle the two NBDs form with respect to AAA2 HBD is also shifted by 1 and 11 degrees in subunits C and F when compared to subunit A. **d** Superimposition of the ADP-Abo1 subunit A (which is representative of all other ADP-Abo1 subunits) onto ATP-Abo1 subunit A and F. ATP-subunit A superimposes well onto ADP-subunit, but ATP-subunit F shows a significant divergence most prominently seen in AAA1 NBD and the linker arm.

determined by AFM matched well with dimensions from our cryo-EM structures, and likely correspond to the C-terminal and AAA2 side of Abo1, as they show hexameric lobes and a central pore that recapitulate pseudo-AFM images simulated using a hard sphere model of the bottom (AAA2) surface of Abo1 but not the top (AAA1) surface (Fig. 6c).

Real-time imaging of wild-type Abo1 in the presence of 2 mM ATP revealed striking symmetry breaking events, where individual blades of the hexameric ring seemed to disappear due to a decrease in the height of a subunit (Fig. 6d and Supplementary Movies 5 and 6). In most asymmetric states, only one blade disappeared from the field of view, but there were also rare transient cases where two blades of the ring disappeared

simultaneously (Supplementary Fig. 14). These HS-AFM results were highly consistent with our cryo-EM structures that show a closed symmetric ring for apo- and ADP states and an open spiral in the ATP-state. Dwell time analysis of ring opening and closing measured a rate of $1.5\,s^{-1}$ for opening and $0.99\,s^{-1}$ ring closing, rates that approximately agree with bulk ATPase rates (Supplementary Fig. 15). Interestingly, the Abo1 Walker B mutant displayed only single symmetry breaking events in the presence of ATP, such that it is "stuck" in the asymmetric spiral conformation (Supplementary Fig. 16; Supplementary Movie 7).

In order to determine how Abo1 utilizes ATP, we tracked the position of ring openings, which indicate which subunit is activated and hydrolyzes ATP. In tracking multiple molecules, we

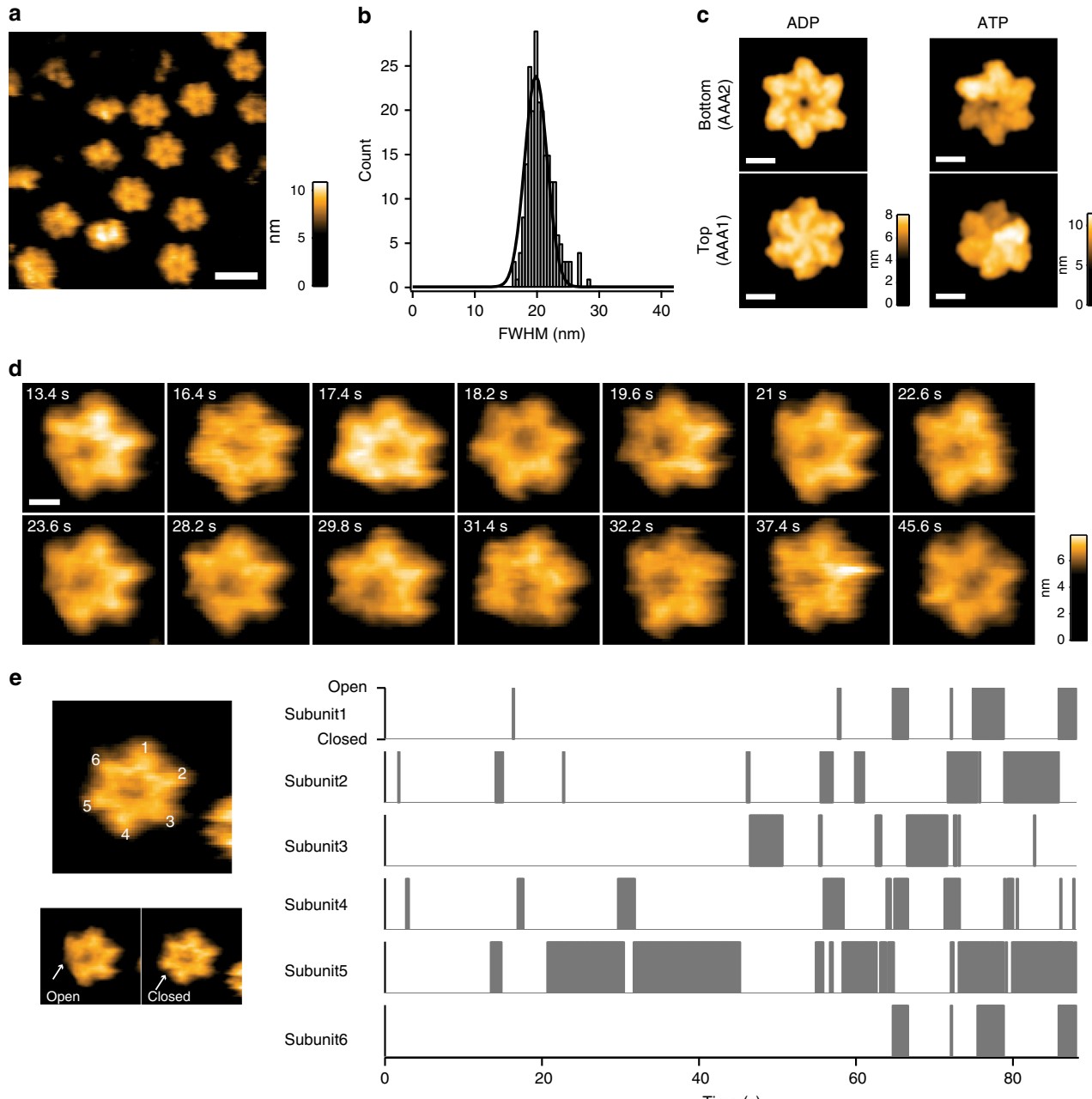

**Fig. 6 Observation of real-time Abo1 conformational change by HS-AFM shows stochastic AAA+ ring symmetry breaking. a** Representative high-speed AFM image of Abo1 hexamer immobilized on an APTES-treated mica surface in the absence of nucleotide. Scale bar = 20 nm. **b** Full width at half-maximum (FWHM) histogram of Abo1 molecules fit to a Gaussian distribution with a mean ± SD of 19.8 ± 2.6 nm. **c** AFM data simulations of Abo1 in different nucleotide states using models from cryo-EM data. Top panels show "bottom view" from the AAA2 domain side, and bottom panels show "top view" from the AAA1 domain side. Scale bar = 5 nm. **d** Snapshots of an Abo1 molecule undergoing conformational change upon ATP addition. Movies were taken at a frame rate of 5 Hz. Scale bar = 5 nm. **e** Analysis of the position of Abo1 ring opening (gray boxes) sorted by subunit according to time shows random subunit activation. Additional examples shown in Supplementary Fig. 17.

found that the ring opening takes place randomly without an ordered sequence (Fig. 6e; Supplementary Fig. 17), indicating that Abo1 subunits are randomly activated. This suggests that at least under basal conditions without added substrate, Abo1 subunits hydrolyze ATP stochastically as proposed for the AAA+ protease ClpX[18].

**Abo1 binds histones with the AAA+ pore and the bromodomain.** Unexpectedly, after building an atomic model into the cryo-EM map of the ATP-Abo1, we discovered a lobe of extra

density in the central pore that could not be accounted for by any part of Abo1 (Fig. 7a). Comparison with the central pore of the ADP- or apo-states showed that this density was unique to the ATP-Abo1 state. We modeled this density as a 14-alanine chain, and examination of interactions between the polyalanine chain and Abo1 revealed that the W345 sidechains of Abo1 form a winding tryptophan staircase around the peptide (Fig. 7b), where the middle subunits (B–E) form direct contacts. The bottom-most tryptophan (subunit A) is completely dislodged from the peptide, while the top-most tryptophan (subunit F) side chain faces away. This type of structure was similar to that seen with other AAA+

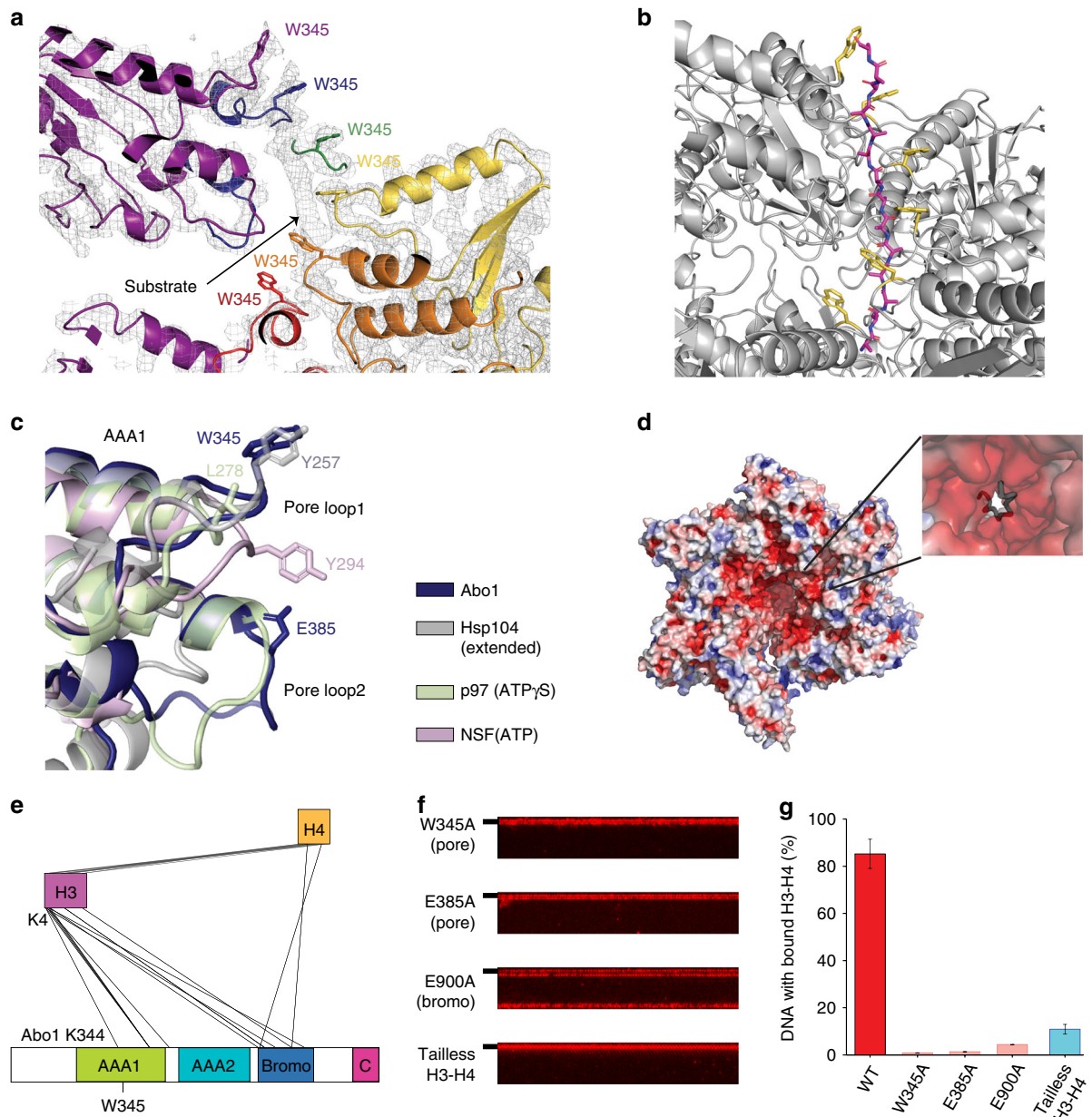

**Fig. 7 Abo1 binds the H3 tail at the central pore to facilitate histone deposition. a** Extra density near the Abo1 central pore region, surrounded by Abo1 W345 tryptophan side chains. **b** Polyalanine model of the substrate peptide (magenta) built into the extra electron density observed in **a**, with W345 side chains colored in yellow. **c** Superposition of Abo1 AAA1 pore loops 1 & 2 with other AAA+ ATPases. Abo1 (dark blue), Hsp104 in the extended state, (gray, PDB ID: 5VYA), p97 in the ATPγS state (light green, PDB ID: 5FTL), and NSF in the ATP state (pink, PDB ID: 3J94) were aligned by AAA1 NBD. Key substrate binding residues on pore loop 1 and the acidic residue of pore loop2, E385, are labeled. **d** Electrostatic surface representation of Abo1 highlighting the negatively charged pore interior contributed by acidic pore loop residues. **e** Depiction of intermolecular crosslinks between Abo1 and histone H3–H4 detected by XL-MS and filtered with a crosslink confidence (LD) score cutoff of 20. The highest LD score intermolecular crosslink, a crosslink between H3 the N-terminus (K4) and Abo1 AAA1 pore loop1 (K344), is labeled. **f** DNA curtain-based H3–H4 deposition assays performed with Abo1 pore loop mutants (W345A and E385A) and a bromodomain mutant (E900A) with wild-type H3–H4, and H3–H4 deposition assays of wild-type Abo1 with N-terminal tail truncated H3–H4. **g** Comparison of H3–H4 loading activity on DNA by quantification of fraction DNA bound with labeled H3–H4. Error bars represent SD from three experiments. The numbers of molecules analyzed for each condition are $n = 244$ for WT, $n = 222$ for W345A, $n = 219$ for E385, $n = 249$ for E900A and $n = 238$ for tailless H3–H4.

ATPases binding substrates such as the tyrosine staircase seen in Hsp104 binding to casein[19] or the tryptophan staircase in Vps4 binding to ESCRT-III[20,21]. Alignment of Abo1 AAA1 with other AAA+ ATPases shows that although the amino acid type is different in different AAA+ ATPases, the position of the key substrate binding residue side chain in pore loop1 is well conserved (Fig. 7c). Based on these observations and similar reports of co-purifying substrates in other AAA+ ATPases[22], we attributed the extra density to the binding of an unknown endogenous substrate(s) such as histones from insect cells that had been inadvertently co-purified with Abo1.

Upon closer examination of Abo1 pore loops, we found two acidic amino acid side chains—E385 of AAA1 pore loop2 and D642 of AAA2 pore loop1—facing the pore interior, thus creating

a negatively charged surface (Fig. 7d). Considering that Abo1 is a histone chaperone, we suspected that some of the extra density in the central pore might correspond to histones or other basic proteins. We were unable to identify any significant hits by mass spectrometry of the cryo-EM Abo1 sample, but when we performed crosslinking-mass spectrometry with Abo1 and H3–H4 (Supplementary Fig. 18), we found many intra- and intermolecular crosslinks (Supplementary Tables 2 and 3). Due to the complexity of the interpretation of intra-molecular crosslinks with the Abo1 homohexamer, we focused on analyzing the intermolecular crosslinks between histones and Abo1, most of which mapped to the Abo1 bromodomain (Fig. 7e and Supplementary Data 1). Interestingly, the highest confidence intermolecular crosslink was between the Abo1 pore loop and the histone H3 N-terminal tail implying that Abo1 might bind histones by the central pore as well as the bromodomain.

To test the potential significance of Abo1 pore—histone tail interactions, we investigated the effect of pore loop mutations and histone tail truncations on the histone loading activity of Abo1. Two Abo1 mutants of conserved pore loop residues (W345A and E385A) displayed very low histone deposition activity in our DNA curtain assay (Fig. 7f, g) despite similar ATPase rates and no significant difference in histone binding compared to wild type (Supplementary Fig. 19). Removing histone N-terminal tails of histone H3–H4 by trypsinization (Fig. 7f, g, and Supplementary Fig. 20) also abolished histone loading on DNA curtains in a similar manner. Therefore, these data suggest that the pore loop residues and histone N-terminal tails are essential for chaperone activity of Abo1.

Because our cryo-EM structure of Abo1 in the apo state shows that the bromodomains form a hexameric ring above the AAA+ ring, we also investigated the contribution of the bromodomains to Abo1 histone loading activity. By mutating E900, a residue that is predicted to be required for histone binding, we found that the E900A bromodomain mutant failed to load histones in the DNA curtain assay (Fig. 7f, g). This implies that both the bromodomains and the pore function in loading histones on DNA by Abo1.

## Discussion

Our studies establish Abo1 as an ATP hydrolysis-dependent histone chaperone, and support in vivo data from S. pombe where Abo1 promotes nucleosome occupancy[9]. Interestingly, previous studies in vivo studies with human (ATAD2) and yeast orthologs (Yta7) proposed that AAA+ histone chaperones evict nucleosomes rather than promote assembly[7,8,23]. Although here we show that Abo1 directly mediates H3–H4 loading onto DNA, it remains to be determined whether there is truly a difference between species, or whether Abo1 can function both to promote histone assembly and disassembly depending on the biological context as has been proposed for other ATP-independent histone chaperones such as FACT[24–26].

Our DNA curtain analyses clearly demonstrate that Abo1 facilitates H3–H4 loading onto DNA in an ATP-dependent manner. MNase treatment of Abo1 assembled H3–H4–DNA complexes show a distinct pattern of protection when compared to CAF-1 assembled H3–H4–DNA complexes. These data suggest that Abo1 loads histones to DNA in a conformation that may be distinct from tetrasomes assembled by CAF-1. It is possible that Abo1 simply loads H3–H4 onto DNA, and requires other factors to assemble proper tetrasomes, or that Abo1-assembled H3–H4 complexes represent an alternate conformation of histone–DNA complexes.

The cryo-EM structures presented in this work reveal that Abo1 assumes a distinct structure that consists of two main parts—a AAA+ ring base and a bromodomain top—that cooperate to bind histones. The bromodomains mimic the 6-fold symmetry of the AAA+ base and assemble into a hexameric ring, thus creating a unique pocket where histones could bind. This arrangement has not been observed with any other bromodomain protein, and is in stark contrast to other histone chaperones which commonly utilize a four-stranded β-structure[26] to bind histones in an ATP-independent manner. Based on our structure and the tighter H3–H4 binding affinity of Abo1 compared to other conventional bromodomains, we speculate that H3–H4 forms a broad binding interface with Abo1 consisting of the inner lining of the bromodomains and the AAA+ ring pore. This is consistent with studies in budding yeast Yta7 (ref. [27]), where the authors found that Yta7 binds histones by regions other than the bromodomains.

Not only is Abo1 distinct as a histone chaperone, but Abo1 is also unique compared to other canonical AAA+ protein structures. While most other AAA+ proteins show dynamic assembly and disassembly of hexameric rings depending on the presence of nucleotides, the Abo1 hexamer is stabilized by structural inserts that mediate tight knob-and-hole packing of individual subunits. Furthermore, AAA2 HBD has a bromodomain inserted with long linkers in between helix 5 and 6 of the HBD, highlighting the versatility of the AAA+ fold.

Analysis of nucleotide-dependent structural changes of Abo1 reveals a striking conformational shift of Abo1 upon ATP hydrolysis that is required for histone loading onto DNA. The spiral-to-ring reorganization is similar to that seen in recent structures of other AAA+ ATPases such as Hsp104 NSF[19,28] and the proteasome[29], although the exact nucleotide state of Abo1 subunits differs. We speculate that such nucleotide-dependent changes in AAA+ ring structure regulate the structure of the bromodomain ring, as the bromodomain density in the apo and ADP states is more distinct compared to the ATP state.

By real-time analysis of Abo1 conformational change, we have directly shown that ATP hydrolysis in Abo1 occurs only in one or two subunits at a time suggesting that ATP hydrolysis in one subunit suppresses ATP hydrolysis in other subunits. We have also shown that in the absence of substrate, there is no ordered sequence of subunit activation, and that the position of symmetry breaking is random. Future studies should elucidate whether this mode of stochastic ATP utilization is maintained even with bound H3–H4 substrate, or whether Abo1 transitions to a sequential mode of ATP hydrolysis analogous to mechanisms proposed for Yme1p[17] and Hsp104 (ref. [28]).

Based on our cryo-EM structures, our current model for Abo1-dependent histone loading on DNA can be summarized as follows. In the ATP-state, Abo1 assumes a closely packed spiral that binds the histone H3 N-terminal through its pore loops and the histone body through its bromodomain. The central pore of Abo1 is ~13 Å in diameter and can accommodate binding of only one histone H3–H4 dimer. When ATP hydrolysis occurs, Abo1 transitions to a planar ring with a larger pore, potentially allowing for remodeling and/or release of histone H3–H4. In this process, we speculate that the energy of ATP hydrolysis might be utilized for exposing the correct histone surfaces and/or to remodel histones to a favorable conformation for DNA deposition. Regardless of the exact role, ATP hydrolysis is essential for Abo1 to load a single H3–H4 dimer onto DNA to initiate nucleosome loading. Alternatively, it is also possible that two Abo1 molecules simultaneously deliver two H3–H4 dimers to DNA in resemblance to other H3–H4 assembly chaperones such as CAF-1 and Rtt106[30,31].

Despite sharing a common structure with other AAA+ ATPases that function as protein disassembly machines, Abo1

performs a different role in facilitating the assembly of higher order structures. This seems to be more akin to the AAA+ ATPase Rvb1/Rvb2-assisted assembly of INO80 (ref. [32]) or the ClpX induced remodeling and activation of ALA synthase[33]. Further structural studies on Abo1-histone complexes should shed light on how the Abo1 AAA+ ATPase uses energy to guide the initial process of nucleosome assembly.

In summary, our work demonstrates the previously unobserved process of ATP-dependent histone deposition and alludes to the possibility that there might be a nucleosome assembly step that requires ATP energy.

## Methods

**Cloning and protein expression of Abo1.** The gene encoding Abo1 aa243–1190 was synthesized (Genscript) and cloned into a pFastBac-HTB vector with an N-terminal 6xHis tag and TEV protease cleavage site. Point mutants of Abo1 were created by inverse PCR with KOD plus polymerase (Toyobo), followed by digestion with DpnI, phosphorylation with T4 phosphonucleotide kinase, and ligation with T4 ligase. Bacmids were prepared from transformed DH10Bac competent cells and transfected into SF9 cells to produce baculovirus. Baculovirus was amplified as suggested by the Bac-to-Bac Baculovirus Expression Kit (Thermo Fisher Scientific). Wild-type and mutant Abo1 proteins were expressed by infection of suspension Sf9 cultures grown in CCM3 media (GE Healthcare) with baculovirus for 43–45 h.

**Purification of Abo1.** Harvested cells were resuspended in lysis buffer (50 mM Tris-HCl pH 8.0, 300 mM NaCl, 5 % glycerol) with a protease inhibitor cocktail (Roche), and lysed by 4 cycles of freezing and thawing. Cell lysates were subsequently cleared by centrifugation at $39,000 \times g$ for 1.5 h, and bound to Ni-NTA resin (Qiagen) pre-equilibrated with lysis buffer for 2 h. Ni-NTA resin was washed with wash buffer (lysis buffer supplemented with 20 mM imidazole) and elution buffer (lysis buffer supplemented with 100 mM imidazole). Eluted fractions were pooled and cut with TEV protease overnight in dialysis buffer (50 mM Tris-HCl pH8.0, 250 mM NaCl, 5% glycerol, 1 mM DTT). Contaminants and uncut His-tagged protein were removed by cycling over Ni-NTA resin 5 times, and further purified by running a 50 mM to 1000 mM NaCl gradient over a HiTrapQ 5 mL column (GE Healthcare). Typical yields were ~50 µg of protein per liter culture of insect cells.

**Fluorescent histone preparation.** Wild-type Xenopus histones were purified by inclusion body preparation as described previously[34]. Lyophilized H4T71C was dissolved in unfolding buffer (20 mM Tris pH 7.5, 6 M Guanidine hydrochloride, 1 mM EDTA, 0.5 mM TCEP), mixed with a 30-fold molar excess of Cy3 or Cy5 maleimide dye (GE Healthcare) and incubated in the dark for 5 h. Excess dye was removed by dialysis and refolded with wild type H3 in refolding buffer (20 mM Tris pH 7.5, 2 M NaCl, 1 mM EDTA, 5 mM 2-mercaptoethanol). Refolded H3–H4 was purified over a Superdex200 increase column in refolding buffer.

**Fluorescence anisotropy.** The fluorescence anisotropy of Cy3-labeled H3–H4 was measured on a SpectraMax M5 Multi-Mode Plate Reader (Molecular Devices) in assay buffer (30 mM HEPES pH 7.5, 200 mM NaCl, 1 % glycerol, 2 mM MgCl₂, 0.01 % NP-40) in a 384-well plate. 0–1 µM of Abo1 was added to 20 nM of Cy3-labeled histone H3–H4. All measurements were performed at room temperature.

**ATPase assays.** The EnzChek Phosphate assay kit (Thermo Fisher Scientific) was used to measure steady state ATPase rates of Abo1. Abo1 was incubated in assay buffer (50 mM Tris pH 8.0, 2 mM MgCl₂, 100 mM NaCl, 1 mM DTT) with or without histone substrates, and 2 mM ATP was added to initiate the reaction at 30 °C. Absorbance at 360 nm was monitored on a Spark multimode microplate reader (Tecan) 5 s after initiating the reaction, and with 10 s intervals thereafter.

**Single-molecule DNA curtain assays.** DNA curtain assays were performed as previously described[13,35]. Briefly, flowcells were prepared by sticking a coverslip onto a fused-silica slide with nanofabricated chromium barriers. A biotinylated lipid bilayer was deposited on the surface by injection of liposomes containing DOPC (1,2-dioleoyl-*sn*-glycero-phosphocholine), 0.5% biotinylated-DOPE (1,2-dipalmitoyl-*sn*-glycero-3-phosphoethanolamine-N-(cap biotinyl)), and 8% mPEG 2000-DOPE (1,2-dioleoyl-*sn*-glycero-3-phosphoethanolamine-N-[methoxy(polyethylene glycol)−2000]) (Avanti Polar Lipids). The surface was further passivated with 0.2 mg/mL BSA. Biotinylated λ-DNA was anchored on biotinylated lipid via streptavidin. The flowcell was connected to a syringe pump fluidic system and placed on a customized prism-type total internal reflection fluorescence (TIRF) microscope that was built on an inverted fluorescence microscope (Eclipse Ti-2, Nikon).

Solid state lasers 488 nm and 637 nm (200 mW, OBIS, Coherent Laser) were used to excite YOYO-1 and Cy5, respectively. The fluorescence from YOYO-1 or Cy5 was collected by a 60x water immersion objective (CFI Plan Apo VC 60XWI, Nikon) imaged on each of two EMCCD cameras (iXON 897, Andor Technology) after passing the optical filter blocking excitation laser.

All histone assembly and disassembly assays were performed at 23 °C in assay buffer (50 mM Tris-HCl pH8.0, 100 mM NaCl, 2 mM MgCl₂, 2 mM DTT, 1.6% glucose and 0.1× gloxy).

For histone assembly assays, lambda phage DNA was first confirmed by the staining with the intercalating agent YOYO-1. To exclude the possibility that YOYO-1 might interfere with histone binding, YOYO-1 was washed out with assay buffer supplemented with 20 mM MgCl₂ and 200 mM NaCl. 5 nM Abo1 and 12.5 nM Cy5–H3–H4 were pre-incubated in buffer supplemented with 1 mM nucleotide cofactor on ice for 15 min, and then injected into the flowcell. Upon the arrival of the maximum amount of proteins to DNA curtains, flow was stopped, and the proteins were incubated with DNA curtains for 15 min. The binding of H3–H4 was examined by resuming flow.

To test Abo1 disassembly activity, we first loaded Cy5–H3–H4 onto DNA using purified yeast CAF-1 (yCAF-1). 5 nM of yCAF-1 was preincubated with 12.5 nM of H3–H4 for 15 min on ice in assay buffer. CAF-1–H3–H4 was injected into the DNA curtain flow cell, incubated for 15 min at room temperature, and washed out with assay buffer for 5 min. Subsequently, 5 nM of Abo1 was injected into the H3–H4-loaded DNA, and movies were taken after 15 min. During all injection and incubation steps, lasers were turned off to prevent Cy5 photobleaching. All DNA curtain movies taken by NIS-Elements (Nikon) were converted to 8-bit TIFF stacks and analyzed in Image J. Kymographs for individual DNA molecules were created, and the fluorescence intensity profile at the tenth frame in each kymograph was obtained. The intensity profile was fitted by multiple Gaussian functions. We analyzed only fluorescence peaks that disappeared when flow was turned off. The peak positions of Gaussians were collected to obtain the binding position distribution of H3–H4.

**MNase digestion assays.** For MNase digestion assays, H3–H4 histones were assembled on 269 bp Widom 601 sequence DNA. 200 nM DNA was mixed with 500 nM H3–H4 dimers and 4 µM yCAF-1 or 4 µM Abo1 and incubated at 23 °C for 1 h. MNase was treated at a concentration of 5 U/µL for 30–90 min in MNase reaction buffer (50 mM Tris-HCl pH 7.9 and 5 mM CaCl₂) at 37 °C, and quenched by addition of 0.3 mg/mL proteinase K and 20 mM EDTA and further incubation for 30 min. Digestion products were extracted with phenol-chloroform, precipitated by ethanol, separated on an 8% non-denaturing gel in 1× TBE buffer, and stained with EtBr.

**Cryo-EM grid preparation.** Prior to sample vitrification, 1.1 mg/mL thawed Abo1 protein was exchanged into a buffer containing 20 mM HEPES pH7.5, 250 mM NaCl, 1 mM DTT, 0.025 % octyl β-D-glucoside, and either 2 mM MgCl₂ or 2 mM MgADP or 2 mM MgATP. 3 µL of Abo1 were loaded onto Quantifoil R1.2/1.3 grids (Quantifoil Micro Tools GmbH) that were glow discharged with 15 mA for 1 min. Grids were blotted for 2–3 s and plunge frozen in liquid ethane cooled by liquid nitrogen using a Vitrobot (Thermo Fisher Scientific).

**Cryo-EM data collection.** Data were collected at three locations: Korea Basic Science Institute (Ochang, Korea) using a Titan Krios 300 keV with a Falcon III direct detector for Abo1-ATP and ADP-Abo1, SciLifeLabs (Stockholm, Sweden) using a Titan Krios 300 keV with a Gatan K2 Summit detector for apo-Abo1, and eBIC at Diamond Light Source (Didcot, UK) using a Titan Krios 300 keV with a Volta phase plate and Gatan K2 Summit detector for apo-Abo1 structure #2. Details of micrographs, exposure, dose, and pixel size are summarized in Supplementary Table 1.

**EM data processing.** Image processing for ATP-Abo1, ADP-Abo1, and apo-Abo1 was performed using cisTEM[36]. Details of the number of selected particles, 2D class averages, initial models, model refinement are summarized in Supplementary Table 1, and the data processing schemes are shown in Supplementary Figs. 4, 9, 10, and 11. All images were motion-corrected with MotionCor2 (ref. [37]) and Ctffind implemented in cisTEM. Particles selection was performed with automated non-template based picking in cisTEM. Several rounds of reference-free 2D classification were performed to remove non-particles and poorly aligning particles, until strong secondary structure features of AAA+ domains were visible in the 2D classes. This subset was used to generate reference-free ab initio models, with no applied symmetry (C1), resulting in an initial 3D model. For ATP-Abo1, 3D classification was performed with 4 classes, using the initial 3D model. The highest resolution class was the most highly populated with 122,644 particles. Using this class we performed another round of 3D classification, selecting the highest resolution class, and an additional round of 2D classification to select the best-aligned classes. Homogeneous refinement and map sharpening resulted in a final model with a resolution of 3.5 Å (ATP-state). Side chains were visible for the AAA+ domains, but only very weak density was visible for the bromodomains and was of poor quality as the local resolution estimates showed in ResMap[38]. For ADP-Abo1 and apo-Abo1, initial processing steps up to 3D classification with 4 classes was the same as for ATP-Abo1. However, for these states, further 3D and/or 2D classification, did not increase the resolution of the structures. Thus, only one round of 3D classification followed by final homogeneous refinement and

sharpening was performed to obtain final models with resolutions of 4.4 Å (ADP state), and 4.3 Å (apo-state) as determined by the gold standard FSC at 0.143 criterion.

Image processing for apo-Abo1 structure #2, which has overall lower resolution than apo-Abo1 structure #1 but improved resolution for the bromodomains, was performed using RELION3.0[39], using a volta-phase plate collected dataset to improve contrast and particle picking. Particle selection was performed using Laplacian-of-Gaussian filtering autopicking. Reference-free 2D classification, initial model generation, and 3D classification was performed as with other datasets, but selection was based more on strong bromodomain density, rather than the overall resolution that includes the AAA+ domain. 3D classification to 4 classes revealed a class that shows stronger bromodomain density than other classes, and subsequent homogeneous 3D refinement and polishing resulted in a final model with a resolution of 6.9 Å. Attempts to further improve the resolution of this model by the application of C6 symmetry, multi-body refinement, and masking were unsuccessful.

**Atomic model building and refinement**. The atomic model of ATP-Abo1 Walker B mutant was built in Coot de novo without the use of any homology templates. The model for chain E, the chain which had the clearest electron density, was built first. Subsequently this model was used as a template to build and refine model other chains. The final model was refined and validated by using cryo-EM validation tools in Phenix.

**Molecular dynamics flexible fitting**. Initial atomic models of apo- and ADP-abo1 were built by constructing six copies of the chain "E" in the de novo structure of ATP-Abo1 Walker B mutant, which has the most complete build achieved at highest resolution among all chains of A-F. The initial models were docked into the apo- and ADP- cryo-EM density maps using the *Fit in Map* module of UCSF Chimera[40], by relaxing each of the six chains independently.

Then, the fits of the atomic models into the apo- and ADP- cryo-EM densities were further optimized by molecular dynamics flexible fitting (MDFF)[16]. We performed two consecutive runs of MDFF for the refinement of the apo- and ADP-Abo1 atomic structures following the default MDFF protocol, carrying out 200,000 MDFF steps for each run. The flexible fittings converged rapidly in all MDFF runs, and the quality of the fits were assessed by running cryo-EM validation tools in Phenix[41].

**HS-AFM imaging**. A laboratory-built high-speed atomic force microscope was used in tapping mode[42]. In brief, a red laser beam was focused on the back side of a small cantilever (Olympus, Tokyo, Japan: BL-AC7DS) and then the reflected laser beam from the cantilever was detected by the optical lever method. The size of the small cantilever is approximately 10 μm × 2 μm × 0.15 μm with a resonant frequency of ~0.6 MHz (in water), a spring constant of ~0.2 N/m, and a quality factor of ~2. An AFM tip was fabricated on the cantilever by using the electron beam deposition (EBD) method and sharpened to be about 2 nm in a radius by ion sputtering in an Ar environment. The free oscillation amplitude of the cantilever was set at ~1 nm. To achieve a small tip-sample loading force, the set point of amplitude for feedback control was approximately 90% of the free amplitude. All HS-AFM measurements were performed at room temperature. A mica surface was cleaved to have a clean and smooth surface for each experiment. Then, the mica surface was chemically modified with 0.05% (3-aminopropyl)triethoxysilane (Shin-Etsu Silicone, Tokyo, Japan). After treatment, a sample droplet of 2 μl was placed on the modified mica surface and incubated for 3 min. The remaining Abo1 molecules were completely removed by rinsing with observation buffer (25 mM HEPES (pH7.5), 250 mM NaCl, 5% glycerol, 1 mM DTT). After the washing, the sample stage was immersed in a chamber with 70 μl of observation buffer.

**HS-AFM image analysis**. A low-pass filter with different cutoff frequencies was applied to all HS-AFM images to reduce spike noises and obtain clear AFM images. Simulation of AFM images was carried out using custom software program based on IgorPro 6 (WaveMetrics, Lake Oswego, USA). The pseudo AFM image was calculated by employing a simple hard sphere model using a conical tip with a radius of 0.5 nm and half cone angle of 10°. After construction of the simulated AFM images, the images were also processed by the low-pass filter with a cutoff wavelength of 2.0 nm.

**Cross-linking mass spectrometry**. To obtain Abo1–H4–H4 complexes, Abo1 and recombinant *X. laevis* H3–H4 were incubated at a 1:3 ratio for 30 min on ice, and then mixed with 1 mM of DSS H12/D12 (Creative Molecules) for 30 min at 37°C with mild shaking on a thermomixer (Eppendorf). The crosslinking reaction was quenched by adding ammonium bicarbonate (Sigma-Aldrich) to a final concentration of 50 mM. The crosslinked mixture was fractionated by a 10–30% sucrose gradient and only fractions corresponding to the Abo1–H3–H4 complex were used for subsequent treatment.

The crosslinked samples were denatured with 8 M urea (Sigma-Aldrich) in the presence of 2.5 mM tris (2-carboxyethyl) phosphine (Sigma-Aldrich). The free cysteine thiols were blocked by adding iodoacetamide (Sigma-Aldrich) to a final concentration of 5 mM. The samples were then sequentially digested with Lys-C

(Wako, Richmond, VA, USA) and trypsin (Promega, Madison, WI, USA) proteases at 1:100 and 1:50 enzyme-to-substrate ratio, respectively. The cross-linked peptides were enriched by SEC (Superdex Peptide 3.2/30, GE Healthcare Life Sciences) and analyzed on an Orbitrap mass spectrometer (Thermo) at the Taplin Mass Spectrometry facility at Harvard. Data analysis was performed using xQuest[43] and a sequence database containing the three target proteins - Abo1, H3, and H4—and a sequence-scrambled decoy. xQuest search results were filtered according to the following criteria: mass error <4 ppm, minimum peptide length = 6 residues, delta score <0.9,% TIC ≥0.1, minimum number of bond cleavages per peptides = 4, and an xQuest LD score cutoff of 20 was selected, corresponding to a false discovery rate of <3%.

**Generation of tailless histones**. 100 μL of Cy5-labeled histones were incubated with 30–50 μL of immobilized trypsin (ThermoFisher Scientific) for 5 min at 4 °C. Immobilized trypsin was removed from histones by passage through an Ultrafree-MC centrifugal filter (Merck) unit.

**Reporting Summary**. Further information on research design is available in the Nature Research Reporting Summary linked to this article.

## Data and materials availability

EM maps are deposited to the Electron Microscopy Data Bank under accession number (EMD-9872 and 9870 for ATP and ADP respectively, and 9871 and 0800 for apo-Abo1 high resolution and apo-Abo1 low resolution respectively). The atomic model of the ATP-state has been deposited to the Protein Data Bank under accession number (PDB ID: 6JQ0, 6JPQ, 6JPU for ATP, ADP, and apo-Abo1, respectively). The source data underlying Figs. 1c, 1d, 2g, 6b, and 7g are provided as a Source Data file.

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

## Acknowledgements
We thank Yumi Shin for excellent technical assistance with baculovirus production and Dr. Taeyang Jung and Dr. Hans Hebert (Karolinska Institutet and The Royal Institute of Technology (KTH)) for help with EM data collection. We would like to thank members of the Song lab for sharing reagents and helpful discussions, and Dr. Hyun Kyu Song for use of a multimode fluorescent plate reader. We acknowledge KBSI (Ochang, Korea) for access to the HR Bio-TEM, the Cryo-EM Swedish National Facility at SciLifeLab (Stockholm, Sweden) for access to the Titan KRIOS, and Diamond (Didcot, UK) for access to the Cryo-EM facilities at the UK national electron bio-imaging centre (eBIC), proposal BI-22985, funded by the Wellcome Trust, MRC and BBSRC. Computing resources were provided by the Global Science experimental Data hub Center (GSDC) at Korea Institute of Science and Technology Information (KISTI). This work was funded by grants (NRF-2016R1A2B3006293, NRF-2016K1A1A2912057, and NRF-2018R1A6A7052113) to J.S., (NRF-2017R1A2B4002213) to J.Y.L., (NRF-2018R1D1A1B07047345) to C.C., a Korea Research Fellowship (NRF-2015H1D3A1066116) to C.C. from The National Research Foundation of Korea, and by the Intelligent Synthetic Biology Center of Global Frontier Project funded by the Ministry of Science and ICT (2011-0031955). J.Y.L. was also supported by research grant IBS-R022-D1-2019-a00. This research was also supported by Joint Research with the Exploratory Research Center on Life and Living Systems (ExCELLS) (ExCELLS program No.18-403 and No. 19–402) and MEXT KAKENHI (Grant number JP19K15412) to H.W.

## Author contributions
C.C. conceived project, designed experiments, performed EM and biochemical experiments, and wrote manuscript, J.J. performed EM and biochemical experiments, H.W. performed and analyzed AFM experiments, T.U. and K.K. supervised AFM experiments, S.J.K. performed MDFF simulations, Y.K. and J.Y.L performed, designed, and analyzed all DNA curtain experiments, J.S. supervised project, designed experiments, performed EM experiments, and edited manuscript.

## Competing interests
The authors declare no competing interests.
