## [Peer Review File · Nature Communications]

Reviewers' comments:

Reviewer #1 (Remarks to the Author):

The ATAD2 family of AAA+ ATPases was first identified in mammalian cells as a co-activator for c-Myc and the androgen receptor (also known as ANCCA). In budding yeast, the Yta7 family member was identified as a regulator of histone gene expression, and more recent work in both budding and fission yeast suggests that ATAD2 family members may regulate nucleosome spacing within transcribed regions by working in concert with the histone chaperone, FACT. Notably, previous work has shown that Yta7 binds directly to histones H3/H4.

In this work, the authors confirm that recombinant ATAD2 forms hexamers (like other AAA+ proteins) and that it can bind to histone H3/H4 tetramers with nanomolar affinity. Although histones do not stimulate the steady state hydrolysis of ATP, the authors present some evidence that histones may promote the initial round of hydrolysis, leading to a phosphate burst. To investigate "nucleosome" assembly, the authors use a DNA curtain assay to follow deposition of H3/H4 tetramers onto DNA. In this assay, deposition requires ATP hydrolysis. The authors then use cryo-EM to solve the structure of ATAD2 hexamers in the absence of nucleotides (Apo state), bound to ADP, or a ATPase dead version (Walker B mutation) in the presence of ATP. Differences between the ADP and ATP bound states suggest that ATP hydrolysis may promote major structural re-arrangements. Finally, the authors identify residues within the "pore" of the ATAD2 hexamer that are key for histone tetramer deposition. Two residues in the pore are acidic, leading the authors to speculate that ATAD2 may bind a histone tail in the pore during the deposition reaction.

The general premise that a AAA+ hexameric ATPase might be an ATP-dependent nucleosome assembly factor is certainly of general interest and would be a novel finding. However, the data shown here do not support this conclusion. First, the authors do not use assays that can monitor nucleosome assembly. The DNA curtain assay only measures deposition of some type of complex that is resistant to flow and does not grossly aggregate the DNA molecule (i.e. it can still be extended under flow). There are standard assays used in the chromatin community for demonstration of nucleosome assembly -- formation of particles that protect ~147 bp of DNA from M_nase digestion or generation of negative supercoils in relaxed plasmid DNA. These types of assays not only demonstrate actual nucleosome assembly, but they can also be performed in a quantitative manner so that the efficiency of assembly can be monitored.

Second, the authors only show deposition of H3/H4 tetramers onto DNA - forming at best tetrasome particles. No assays are used to actually monitor nucleosome assembly.

The ATP-dependence of the deposition reaction is interesting, but it seems likely that energy is used for assembly of a functional ATAD2 ring structure, rather than deposition per se. Indeed, the cryo-EM work suggests that ATP hydrolysis controls ring assembly/dynamics. The authors could incubate ATAD2 with ATP, remove ATP, and then perform histone binding and deposition to isolate the ATP-dependent step. Note that such a model would be similar to how the Ruvb1/Ruvb2 AAA+ ring within the INO80 remodeling enzyme uses ATP to control assembly of this complex.

The authors present a model for how ATAD2 binds to histones and promotes deposition. This model has two simple tests - the bromodomain should be essential and ATAD2 should not be able to deposit H3/H4 tetramers that lack their N-terminal tail domains. These are simple experiments that would strengthen this story.

Other points:

1. In figure 1E, the authors show initial Pi burst by ATAD2. No units are given on the y-axis, nor is it clear if data were normalized to reactions that lacked histones. Furthermore, phosphate burst experiments are normally performed in a stopped-flow apparatus so that the burst kinetics can be

quantified. This current assay is a commercial kit that does not allow accurate measurements. Based on this one assay, I do not think that the authors can conclude that ATAD2 is a "histone H3-H4 regulated ATPase".

2. Figure 2. It is quite odd that addition of H3/H4 tetramers to the DNA curtains do not yield complexes. Histones will aggregate readily on DNA - they do not remain in solution if DNA is present. Perhaps the histones are bound to DNA, but these aggregates do not extend under flow?

3. The supplemental figure showing deposition by CAF1 is not strong. The CAF1 reaction should be very effective at promoting nucleosome or tetrasome assembly, but few particles appear to be formed. This again points to the non-quantitative nature of this assay.

4. The cryo-EM structures appear to be of high quality, but I question whether the authors can directly compare the wildtype and ATPase dead hexamers, concluding that nucleotide hydrolysis is the only difference. Perhaps the authors could show the ATPase dead form in the apo and/or ADP state as well?

5. The authors' model suggests a major role of the ATAD2 bromodomain in histone recognition. However, bromodomains show specificity for acetylated lysines on histones, not un-modified histones. Furthermore, bromodomains typically bind to histone tails with micromolar affinity, not nanomolar. Unless the ATAD2 bromodomain is unique, the properties of bromodomains do not seem consistent with the model.

Reviewer #2 (Remarks to the Author):

General comments: Cho et al. provide exciting data on the structure of the yeast ATAD2 homolog, Abo1, at atomic resolution using Cryo-EM. The main achievement is the visualization of the transition of Abo1 from an asymmetric spiral (ATP state) to a symmetric ring (ADP and apo-states). The conformational changes undergone in this process are analyzed by comparison of the atomic structures and they are also observed by HS-AFM, showing a random aperture of the ring. Using DNA curtain assays they demonstrate that Abo1 interacts with histone H3/H4 allowing its assembly onto DNA, defining its role as a histone assembly factor.

Major concerns:

1. Although the interaction between Abo1 and histone H3/H4 is clear, a cryo-EM structure of the complex Abo1-H3/H4 would be very relevant and provide high-quality structural information regarding this interaction. Without that structure, some of the conclusions exposed in the final model are quite speculative and are not properly supported by the experimental evidence. Is there any reason why this reconstruction has not been attempted?

2. The cryo-EM images presented in the supplementary figures 9 and 10 show some 2D averages in which the bromodomain is clearly defined. I wonder if the authors have processed this information to its full potential. By the use of different classification techniques and masks, I believe they should be able to obtain a structure of the bromodomain at least to a medium resolution. I suggest as well to analyze the classes obtained in the processing of ATP-Abo1, there is a class with 30% of the particles that might correspond to an intermediate state and provide interesting information on the transition between states.

Specific comments:

1. There are some issues regarding incorrect writing, figure citing and figure legends. I suggest a more careful revision of the whole manuscript. There is even an incomplete sentence in the discussion. Please revise.

2. A more detailed argument why the full-length version of Abo1 has not been used is required.
3. No Cryo-EM structures of Abo1-Walker B mutant are shown in Figure 3. They should be shown together with the atomic model. I strongly recommend removing the bromodomain cartoons. Related with this figure, it would be nice to give more details about the dimensions of this 3DR, scale bars and a brief description of the Cryo-EM structure of Abo1 in the main text.
4. Figure 4: bottom views are missing (indicated in the legend but not shown) and I suggest presenting as well the Apo-Abo1 3DR. Regarding ADP-Abo1 and Apo-Abo1 3DRs, it seems that individual subunits in both states do not show any significant differences in their structures, supporting the idea that both states are symmetric. Have the authors thought to apply six-fold symmetry in order to improve the resolution of the AAA domains and obtain a better definition of the bromodomains?
5. Figure 5: providing the angle measurements and the hinges used to show the movement of the subunits would allow a better understanding of the transition.
6. Regarding the real-time ATP-dependent structural changes of Abo1 studied by HS-AFM I wonder if the subunit activation is really random. It seems that subunit 5 in Figure 6e stays open much longer than the rest of subunits. I observed similar behavior in the examples of supplementary Figure 13. Have the authors analyzed the possibility that one of the subunits is actually favored to initiate activation?
7. Related with the substrate binding, I would suggest another representation for the substrate in order to identify clearly the density designated to the substrate. No clear extra mass can be observed in figure 7a as it is. Besides, without any confirmation by mass spectrometry the statement "the extra density in the central pore corresponded to a part of endogenous histone H3-H4 from insect cells" needs to be toned down. Wasn't there any protein at all detected by MALDI-MS in addition to Abo1 that might account for that mass?
8. More information about H3-H4 histones used in the cross-linking experiments would be required either in the main text or methods sections. Why did the authors purify the histones from *Xenopus*? Have they attempted purification from yeast?
9. Having such clear crosslinks between H3 and AAA1, AAA2 and the bromodomains and, to a lesser extent, H4 and the bromodomains, I would like to know why the reconstruction of this complex has not been attempted.

Specific comments related to Cryo-EM data:

1. In my opinion, the strongest results from the paper are those related to EM. For that reason, I recommend providing more details regarding image processing and 3D reconstruction. Information concerning the selection of 2D and 3D classes or the generation of initial models is missing and would help to evaluate the quality of the processing.
2. Again, I am concerned that image processing has not been optimized to its maximum potential. Some 2D averages in Apo and ADP states should allow a much better definition of the bromodomains. The use of tools from Relion package like Multi-body, Subtraction or the application of masks can help to define the most flexible regions. Besides, sharpening tools like LocalScale and LocalDeblur can help in the interpretations of 3D reconstructions with areas of different resolution.
3. Also mentioned before, the application of 6-fold symmetry to the Apo and ADP reconstructions should allow obtaining a better definition of the different domains.

Specific comments related to Model Building:

1. In the validation reports for the different structures, it seems that the near full-length mutant reported in the main text was used only for the ATP state (837 residues), whereas a sequence with 1190 residues, that must account for the full-length protein, has been used for Apo and ADP states. Can the authors explain the reason for the use of the different versions of the protein?
2. I believe that the model building can be improved using manual adjustment and real-space refinement with Coot to increase the quality of the fitting. This way outliers and angle deviation values should be reduced to meet the standards required for this kind of structures. This is strongly recommended for Apo and ADP structures. I am concerned about the low correlation coefficient (0.6) between the map and the model of the Apo state. These parameters should be improved.

Reviewer #3 (Remarks to the Author):

The manuscript by Cho, Jang et al examines loading of histone H3/H4 dimers by the AAA+ AtPase nucleosome chaperone Abo1 in *S. pombe* by a variety of structural and single molecule methods. Single particle EM reconstructions of several nucleotide bound states of the hexameric Abo1 complex have been generated at 3.5-4.5Å. This seems to be the first structure of an AAA+ atPase from the class that also contains a bromodomain and is therefore thought to be a nucleosome assembly/dissassembly chaperone. Although the bromodomain remains unmodeled here, the atomistic model of the ATPase domains show features that are distinct from other AAA+ atPases and suggest a model of binding the histone tails in the central pore of the ring.

Generally, this paper seems methodologically solid and represents an advance in this field (which admittedly is not my area of expertise). I do have some concerns/questions.

- 1) Reference 9 reports high homology between Abo1 and Abo2 and they are found to interact via synthetic lethality assay. Rvb1/Rvb2 are AAA+ atPases that in yeast form alternating Rvb1/Rvb2 hexamers. Is the EM reconstruction at sufficient resolution that it conclusively rules out such an arrangement with Abo1/Abo2?
- 2) All of the histone binding assays and structures seem to be done with unmodified histones. The presence of the bromodomain suggests that the endogenous substrates would be acetylated somewhere along the tails. The paper would be stronger if it examined binding of Abo1 to acetylated histone substrates.
- 3) The inability to identify the central pore substrate (figure 7) is disappointing. The CLMS results and the analysis of the W345A and E385A mutant doesn't convince me that histone tails bind directly in the channel. CLMS can be measuring multiple configuration/conformation states of the complex simultaneously and it's not clear how the bromodomain and the AAA1 domain can bind the same molecule. The fact that the crosslink between H3-K4 and Abo1-K344 has the highest confidence score does not mean anything except that this particular crosslinked peptide pair ionizes and fragments well. There is only a single crosslink implicating this residue, whereas a number of crosslinks are found between the H3-tail and the C-term side of AAA1 as well as the bromodomain.

The mutation analysis indicates that the mutants bind H3-H4 and hydrolyse ATP at same level as wt while not assembling H3-H4 onto DNA. It could be that some histone tails must bind in the central pore as part of the mechanistic cycle, or it could just be that these residues are necessary for dna loading for some other mechanistic reason.

4) The CLMS data is not sufficiently described. The sample for CLMS is prepared with quite a high excess of histone (it is not clear if the molar excess of 1:3 is to Abo1 monomer or hexamer in the methods). Also, no FDR is reported for the CLMS dataset and since the search is against a very small sequence database it is prone to false positives as the distribution of decoy crosslink hits can not be assessed very well. Overall numbers of crosslinks found and distribution between proteins is not reported.

It is inconceivable that there were no Abo1-Abo1 crosslinks discovered. The fidelity of the CLMS data should be assessed by measuring the distance distribution of the measurable crosslinks against the reported structural model. While MS cannot easily distinguish which pair of protomers produce a given crosslink, it is common practice in this field to measure all 5-possible Lys-Lys distances and reported the most likely (shortest distance). Hence the violation rate of the dataset can be assessed. Additionally, the crosslinking data should provide more information about the localization of the bromodomain.

In short, I believe the paper is fairly high quality but I find the substrate binding model (Fig 7C) not particularly likely. The paper could be improved by further exploration of the substrate characteristics in terms of histone acetylation, stoichiometry of histone binding, and by trying to identify a DNA binding region of the complex. The CLMS data is poorly reported, and seemingly relevant crosslinks are hidden from discussion when they contain other relevant information to the paper.

Reviewer #4 (Remarks to the Author):

In this manuscript, Cho and colleagues report a series of CryoEM structures of the Abo1 AAA+ ATPase. The ATP-saturated structure of the Walker B mutant shows a broken ring/washer hexamer with ATP molecules located at the interfaces between the protomer (except for the broken interface). Resolution of CryoEM map for this complex allowed the authors to build the structure of the ATPase part (sans bromodomain) de novo. The ADP-bound complex was visualized at lower resolution, but sufficient to reveal a more symmetric, closed ring structure. A number of features conserved among AAA+ ATPase and those unique to Abo1 are discussed. These structures will be important to the AAA+ ATPase field and will advance our understanding of the histone chaperones. Most interestingly, the high speed AFM imaging of the wild type Abo1 in the presence of ATP confirmed that the hexamer transitions between the closed ring and the open washer conformations with interface breaking at random subunit, which the authors argue corresponds to the ATP hydrolysis.

These are very impressive data, but it would be very informative if the authors show representative images of the open complexes where more than one subunit is off the mica plain. The time series in Fig 6E and in Supplemental Figure 13 suggest that the ring opening occasionally occurs simultaneously at multiple subunits, and even at non-adjacent subunits (e.g. 2,3,5 in Fig 6E). Without an AFM frame, it is difficult to imagine this event without the hexamer completely dissociating, as the interface in the CryoEM structure seems to be broken across both AAA1 and AAA2 domains.

Another point here. From the time series of the ring openings observed by AFM, the authors should be able to estimate a k_{cat} for ATP hydrolysis. Does it correspond to the ATPase activity measured in solution? A good correspondence here would straighten the authors' model where the opening of the ring corresponds to the ATP hydrolysis.

In addition to structural information, the authors also report functional studies, as it has been unclear what the Abo1 function is with respect to histone H3-H4. The authors show that Abo1 interacts with H3-H4 and promotes its deposition on DNA in an ATP-dependent manner. The latter

was investigated using single-molecule DNA curtain technique. This is an important piece of data in the manuscript and qualitatively, the conclusions seems correct – Abo1 loads, but does not unload H3-H4 onto the DNA and the ATP hydrolysis is indeed important. Only minimal quantification of the single-molecule experiments, however, is presented and a lot of important information is missing in the methods.

1. How many molecules were analyzed in each experiment?
2. Why particular concentrations of Abo1 and H3-H4 were selected? Considering a K_d of about 23 nM, only a small fraction of Abo1 will be in complex with H3-H4. Is it expected to work catalytically?
3. Some non-specific binding of H3-H4 to the surface can be observed (Fig. 2D “Flow off” panel). How was this non-specific binding distinguished from the actual binding to DNA?
4. Is there any specificity to the H3-H4 deposition, or is it completely random?

Minor point. The manuscript can benefit from some editing. Many sentences throughout the manuscript are difficult to understand, which detracts from the beautiful data the authors try to communicate.

Reviewers' comments:

Reviewer #1 (Remarks to the Author):

The ATAD2 family of AAA+ ATPases was first identified in mammalian cells as a co-activator for c-Myc and the androgen receptor (also known as ANCCA). In budding yeast, the Yta7 family member was identified as a regulator of histone gene expression, and more recent work in both budding and fission yeast suggests that ATAD2 family members may regulate nucleosome spacing within transcribed regions by working in concert with the histone chaperone, FACT. Notably, previous work has shown that Yta7 binds directly to histones H3/H4.

In this work, the authors confirm that recombinant ATAD2 forms hexamers (like other AAA+ proteins) and that it can bind to histone H3/H4 tetramers with nanomolar affinity. Although histones do not stimulate the steady state hydrolysis of ATP, the authors present some evidence that histones may promote the initial round of hydrolysis, leading to a phosphate burst. To investigate "nucleosome" assembly, the authors use a DNA curtain assay to follow deposition of H3/H4 tetramers onto DNA. In this assay, deposition requires ATP hydrolysis. The authors then use cryo-EM to solve the structure of ATAD2 hexamers in the absence of nucleotides (Apo state), bound to ADP, or a ATPase dead version (Walker B mutation) in the presence of ATP. Differences between the ADP and ATP bound states suggest that ATP hydrolysis may promote major structural re-arrangements. Finally, the authors identify residues within the "pore" of the ATAD2 hexamer that are key for histone tetramer deposition. Two residues in the pore are acidic, leading the authors to speculate that ATAD2 may bind a histone tail in the pore during the deposition reaction.

The general premise that a AAA+ hexameric ATPase might be an ATP-dependent nucleosome assembly factor is certainly of general interest and would be a novel finding. However, the data shown here do not support this conclusion.

1. First, the authors do not use assays that can monitor nucleosome assembly. The DNA curtain assay only measures deposition of some type of complex that is resistant to flow and does not grossly aggregate the DNA molecule (i.e. it can still be extended under flow). There are standard assays used in the chromatin community for demonstration of nucleosome assembly -- formation of particles that protect ~147 bp of DNA from Mnase digestion or generation of negative supercoils in relaxed plasmid DNA. These types of assays not only demonstrate actual nucleosome assembly, but they can also be performed in a quantitative manner so that the efficiency of assembly can be monitored.

→ According to the reviewer's suggestion, we performed MNase protection assays to probe the conformation of H3-H4 deposited on DNA (Supplementary Fig. 2b). Although the smeared protection patterns of Abo1-H3/H4 hindered us from performing quantitative analysis, our MNase digestion assay with Abo1-H3/H4 shows a distinct pattern of DNA protection, compared with the pattern observed in the presence of CAF-1+H3/H4, which conferred partial protection of ~70-80bp DNA fragments, as expected for tetrasome arrays (Feng Dong and K. E. van Holde PNAS 88, 10596-10600 (1991) & Francesca Mattioli et al. eLife 6, e22799 (2017)). Specifically, two groups of protected fragments at shorter (<50bp) and longer (140-150 bp) lengths appear in the presence of Abo1+H3-H4. The shorter protected fragments seem to result from protection by Abo1 alone, whereas the longer protected fragments appear only in presence of Abo1 and H3H4.

At this point, we are unable to discern the exact structural characteristics of the longer protected fragments. However, these data suggest that Abo1 delivers histones to DNA in a conformation that may be distinct from tetrasomes that are assembled by CAF-1. One possibility is that Abo1

simply loads H3-H4 onto DNA, and requires other factors to assemble proper tetrasomes, and somehow H3-H4 together with Abo1 may protect a large region of DNA, which corresponds to the longer protected fragment. Despite these data, our DNA curtain analysis clearly demonstrates that histone H3-H4 is deposited onto DNA in an Abo1- and ATP-dependent manner.

As the reviewer pointed out, Abo1 alone may not be able to generate tetrasomes on its own. Therefore, we have revised the manuscript to use the term histone “deposition” instead of “assembly” throughout the manuscript.

2. Second, the authors only show deposition of H3/H4 tetramers onto DNA - forming at best tetrasome particles. No assays are used to actually monitor nucleosome assembly.

→ We acknowledge that we are only observing the initial step of nucleosome assembly, H3-H4 binding to DNA, in our assays. Considering this, we have revised the manuscript to accurately reflect the fact that we are observing H3H4 deposition onto DNA, and not nucleosome assembly per se.

The ATP-dependence of the deposition reaction is interesting, but it seems likely that energy is used for assembly of a functional ATAD2 ring structure, rather than deposition per se.

→ In many other AAA ATPases, ATP is required for forming hexameric ring structure. However, we have clearly shown in the cryo-EM structures (Fig.) and gel-filtration analysis (Fig. 1b) that Abo1 assembles into hexameric rings even in the absence of any nucleotide, and that multiple structural features stabilize the Abo1 ring such that it does not dynamically disassemble and reassemble in different nucleotide states.

In addition, our DNA curtain assays show that localization of histone H3-H4 onto DNA curtains is dependent on the presence of both Abo1 and ATP. The specific localization of H3-H4 onto DNA curtains does not occur in the presence of ATP analogs, nor with the ATP-hydrolysis deficient Abo1 mutant, showing that this process is Abo1 ATP-hydrolysis-dependent.

There are many possibilities for why ATP energy is required – it might be required for Abo1 to rearrange into a conformation that is competent for H3H4 release, or for Abo1 to reposition H3H4 into a conformation that is favorable for DNA binding. At this point we are unable to distinguish between these and many other possibilities, and we will have to reserve this question until we have more structural information on the Abo-histone H3H4 complex, as discussed below.

3. Indeed, the cryo-EM work suggests that ATP hydrolysis controls ring assembly/dynamics.

The authors could incubate ATAD2 with ATP, remove ATP, and then perform histone binding and deposition to isolate the ATP-dependent step. Note that such a model would be similar to how the Ruvb1/Ruvb2 AAA+ ring within the INO80 remodeling enzyme uses ATP to control assembly of this complex.

→ In contrast to the Ruvb1/Ruvb2 complex that dynamically switches between different oligomeric states under different conditions, Abo1 maintains a stable hexamer with two AAA domains throughout its ATPase cycle. Although Ruvb1/Ruvb2 is another AAA+ histone chaperone akin to Abo1, we predict that the molecular mechanism of ATAD2 likely differs from Ruvb1/Ruvb2 based on the differences in the structures and biochemical properties.

We have shown in the DNA curtain assays that ATAD2 ATP hydrolysis is required for histone deposition, and that nucleotide binding does not suffice for histone deposition to occur. In our current DNA curtain assay, we are not able to distinguish the individual steps of Abo1-histone binding and histone deposition. In order to fully dissect the ATP-dependent step in Abo1-histone deposition one would have to perform sophisticated single-molecule fluorescence or FRET experiments with labeled Abo1 to simultaneously monitor the opening/closing of the Abo1 hexamer and histone deposition. However, we believe that this is beyond the scope of the

current manuscript describing the first cryo-EM structures of Abo1, and should be reserved for future work dissecting the structural basis of Abo1 binding and DNA deposition.

4. The authors present a model for how ATAD2 binds to histones and promotes deposition. This model has two simple tests - the bromodomain should be essential and ATAD2 should not be able to deposit H3/H4 tetramers that lack their N-terminal tail domains. These are simple experiments that would strengthen this story.

→ We thank the reviewer for his/her insightful comments. We have performed the suggested experiments with the data shown in Figure 7f and Supplementary Fig. 20. First, to disrupt bromodomain function, we introduced a E900A mutation (corresponding to the N1064 residue in ATAD2 as shown in Morozumi et al (2015)) that is predicted to be a critical amino acid in the histone binding pocket of the Abo1 bromodomain. The bromodomain mutant E900A disrupted DNA deposition of H3-H4 in the DNA curtain assay suggesting that histone binding by the Abo1 bromodomain is required for Abo1 activity. Second, we performed the DNA curtain assay with H3H4 molecules that lack the H3 and H4 N-terminus by tryptic digestion of histone N-terminal tails. This mutant also disrupts histone deposition in the DNA curtain assay, suggesting that the histone N-terminal tails are required for Abo1-dependent DNA deposition.

Other points:

5. In figure 1E, the authors show initial Pi burst by ATAD2. No units are given on the y-axis, nor is it clear if data were normalized to reactions that lacked histones. Furthermore, phosphate burst experiments are normally performed in a stopped-flow apparatus so that the burst kinetics can be quantified. This current assay is a commercial kit that does not allow accurate measurements. Based on this one assay, I do not think that the authors can conclude that ATAD2 is a "histone H3-H4 regulated ATPase".

→ The accuracy and sensitivity of the commercial ATPase kit used in the paper has been shown to be comparable to other non-commercial assays and has been widely used to measure steady state ATPase and GTPase rates. (Mogami et al (2007), Furuta et al (2008), and O'Donnell et al (2017)). However, we agree in order to accurately measure the "phosphate burst", pre-steady state stopped flow kinetics experiments must be performed, and the commercial kit we have used is inappropriate for this purpose. We have thus excluded the data on phosphate bursts in the revised manuscript.

6. Figure 2. It is quite odd that addition of H3/H4 tetramers to the DNA curtains do not yield complexes. Histones will aggregate readily on DNA - they do not remain in solution if DNA is present. Perhaps the histones are bound to DNA, but these aggregates do not extend under flow?

→ As the reviewer mentioned, histones will usually aggregate on DNA in bulk assays. But in our DNA curtain assay, we optimized the H3-H4 concentration to a very low concentration (12.5nM) to minimize spontaneous binding and aggregation of H3-H4 on DNA. Therefore, we do not observe any shortening of DNA molecules by histone deposition on the DNA curtain.

7. The supplemental figure showing deposition by CAF1 is not strong. The CAF1 reaction should be very effective at promoting nucleosome or tetrasome assembly, but few particles appear to be formed. This again points to the non-quantitative nature of this assay.

→ The movies shown in the supplementary material give a qualitative view of histone assembly on DNA curtains, but do not provide direct information on the histone loading efficiency.

By detailed analysis of DNA curtain movies, we have now added statistical information on the position of histone molecules localized on DNA curtains in Supplementary Fig. 2A, that should provide a better understanding of how histones are positioned on DNA curtains. We also include quantification of the percent of DNA molecules with histone H3-H4 bound in Supplementary Fig. 3. This shows that CAF-1 mediated H3H4 deposition is sufficient to observe the disassembly by Abo1, if any, and that adding Abo1 to CAF-1 assembled tetrasomes does not significantly change the pattern of H3H4 deposition.

8. The cryo-EM structures appear to be of high quality, but I question whether the authors can directly compare the wildtype and ATPase dead hexamers, concluding that nucleotide hydrolysis is the only difference. Perhaps the authors could show the ATPase dead form in the apo and/or ADP state as well?

→ The use of a Walker B mutant in complex with ATP is a very commonly employed strategy in studies of AAA+ ATPases to capture the structure of a pre-ATP hydrolysis intermediate (Puchades et al, (2017), Chen et al (2010), and Lee et al (2012)). In many cases, the structure of wild-type AAA+ ATPases mixed with nucleotide analogs assume similar structures to Walker B mutants. Also, as shown in our HS-AFM data (Supplementary Fig. 16), we observe that Abo1 Walker B mutant molecules can initially assume structures that are similar to wild-type Abo1, confirming that Walker-B mutants are reflective of wild-type Abo1 structure.

9. The authors' model suggests a major role of the ATAD2 bromodomain in histone recognition. However, bromodomains show specificity for acetylated lysines on histones, not un-modified histones. Furthermore, bromodomains typically bind to histone tails with micromolar affinity, not nanomolar. Unless the ATAD2 bromodomain is unique, the properties of bromodomains do not seem consistent with the model.

> As the reviewer has correctly pointed out, bromodomains in general specifically recognize acetylated lysines, not unmodified histones. In these studies, we have used unmodified histones, and we acknowledge that the use of modified histones could potentially increase the selectivity and strength of ATAD2-histone interactions. However, the high affinity of Abo1 even for unmodified histones observed in our study is not contradictory to our model, but rather highlights the uniqueness and complexity of ATAD2 bromodomain organization due to its association with AAA+ domains. To explain the non-canonical nature of the ATAD2/Abo1 bromodomain in more detail,

1) There are 6 bromodomains in ATAD2 that are forced into hexameric arrangement (new Supplementary Fig. 12) due to the hexameric ring structure of the AAA+ base of ATAD2. This arrangement would increase the affinity for histones compared to other bromodomain-containing proteins that usually contain only 1 bromodomain per molecule.

2) Judging from our crosslinking-MS data, ATAD2 interacts with histones not only by interactions of the bromodomain with the histone tails, but also by interactions between the bromodomain and histone body, and interactions between the AAA1 domain and histones. This is consistent with the findings of (Koo et al, (2015)) where the authors measured a micromolar affinity of histone peptides for the isolated ATAD2 bromodomain, but nanomolar affinity of histone peptides for a longer construct of ATAD2 that encompasses the AAA+ and bromodomains. This might also be related to the fact that the ATAD2 bromodomain has a highly negative electrostatic surface potential (Filippakopoulos et al (2012)), and could potentially promote electrostatic interactions between the histone body and ATAD2. Furthermore, this is also consistent with the findings of Gradolatto et al (2009), where the authors found that Yta7 contains histone binding regions outside of the bromodomain.

3) There is still some controversy as to which acetyllysine(s) are the major target(s) of ANCCA. Koo et al (2015) and Morozumi et al (2015) identified H4K5ac and H4K12ac as major targets of

the ATAD2 bromodomain, while Revenko et al (2010) found that H3K14ac preferentially binds the ATAD2 bromodomain. In the case of the budding yeast ortholog, Yta7, Gradolatto et al (2009) found that bromodomain binding to histones was unaffected by posttranslational modifications.

4) Adding to this complexity, comparison of yeast Abo1 and human ATAD2 amino acid sequence shows that Abo1 diverges significantly from ATAD2 and other bromodomains. Secondary structure predictions predict that Abo1 adopts the same overall 4-helical bundle fold as other bromodomains, but Abo1 notably diverges in key residues that have been shown to be essential for acetyllysine recognition such as 1021Y and 1064N of ATAD2. In addition, the effects of histone acetylation on Abo1 activity have not been tested, and the target of the Abo1 bromodomain have not been identified.

These observations together establish that ATAD2/Abo1 diverge in character from conventional bromodomain proteins, and thus require full in-depth studies of their characteristics. We plan to pursue future studies relating to the identification of Abo1-specific acetylated histone peptides and the effect of histone acetylation on Abo1 function, but at this point, we believe this is beyond the scope of the current study.

Reviewer #2 (Remarks to the Author):

General comments: Cho et al. provide exciting data on the structure of the yeast ATAD2 homolog, Abo1, at atomic resolution using Cryo-EM. The main achievement is the visualization of the transition of Abo1 from an asymmetric spiral (ATP state) to a symmetric ring (ADP and apo-states). The conformational changes undergone in this process are analyzed by comparison of the atomic structures and they are also observed by HS-AFM, showing a random aperture of the ring. Using DNA curtain assays they demonstrate that Abo1 interacts with histone H3/H4 allowing its assembly onto DNA, defining its role as a histone assembly factor.

Major concerns:

1. Although the interaction between Abo1 and histone H3/H4 is clear, a cryo-EM structure of the complex Abo1-H3/H4 would be very relevant and provide high-quality structural information regarding this interaction. Without that structure, some of the conclusions exposed in the final model are quite speculative and are not properly supported by the experimental evidence. Is there any reason why this reconstruction has not been attempted?

→ We agree that a cryo-EM structure of Abo1 in complex with histones would greatly enhance our mechanistic understanding of Abo1 and provide stronger support for our model. We are pursuing this line of study; however, obtaining a structure of a AAA+ protein in complex with its substrate is a challenging feat as has been exemplified even for relatively amenable AAA+ proteins such as bacterial proteases and heat shock proteins.

We would like to point out that our study presents the first structures of the ATAD2 family of proteins, and that the purification and structural characterization of Abo1 in itself has been a major scientific and technical advance. We plan to continue to dissect the molecular details of the Abo1-H3H4 interaction, but we believe that this will require another technical breakthrough that is beyond the scope of this study.

In terms of our model, we acknowledge that we lack details in how Abo1 binds and releases histone H3H4 substrates, and have accordingly removed the model figure. However, we have provided evidence to support our claims that 1) Abo1 interacts with H3H4 both through its bromodomains and AAA+ domains and that 2) the AAA+ pore interaction with the histone N-terminal tail is required for H3-H4 deposition onto DNA. In the revised manuscript, we include additional data showing that the bromodomain is critical for the function of Abo1 shown by DNA curtain assay using a bromodomain mutant (Abo1 E900A). We also show that the H3-H4 N-terminal tails are required for H3-H4 loading onto DNA.

2. The cryo-EM images presented in the supplementary figures 9 and 10 show some 2D averages in which the bromodomain is clearly defined. I wonder if the authors have processed this information to its full potential. By the use of different classification techniques and masks, I believe they should be able to obtain a structure of the bromodomain at least to a medium resolution. I suggest as well to analyze the classes obtained in the processing of ATP-Abo1, there is a class with 30% of the particles that might correspond to an intermediate state and provide interesting information on the transition between states.

→ We were unable to obtain an improved model of ATP-Abo1 despite attempts with multibody refinement, masking, etc. Regarding the reviewer's comments on processing of the ATP-Abo1 structure, although 30% of the particles in the Abo1-ATP bound state classified as a different class during 3D classification, further processing of this class revealed an overall structure that is similar to the high-resolution structure of Abo1-ATP bound state, but gave lower resolution. As the reviewer mentioned, it is possible that 30% of the particles result from an alternative conformation of Abo1, but at the moment, we are unable to gain a higher resolution

reconstruction of this class, and are unable to distinguish this state from our current Abo1-ATP structure. Instead, we have performed further processing and obtained an improved structure of the apo state included in the revised manuscript. For this, we collected a new data set for Abo1 in the apo state using a Volta phase plate. The new cryo-EM structure clearly shows the bromodomain above the AAA+ ring forming another hexameric ring (Supplementary Fig. 12) . We included these new data in the revised manuscript.

Specific comments:

1. There are some issues regarding incorrect writing, figure citing and figure legends. I suggest a more careful revision of the whole manuscript. There is even an incomplete sentence in the discussion. Please revise.

→ We have reviewed the manuscript in detail and have made changes in incorrect wording and figure legends.

2. A more detailed argument why the full-length version of Abo1 has not been used is required.

→ The full-length version of Abo1 could not be generated in our recombinant expression system, and only by removal of the N-terminus were we able to obtain sufficient quantities of Abo1 for structural analysis. We have included comments regarding the generation of our construct in the Methods section.

3. No Cryo-EM structures of Abo1-Walker B mutant are shown in Figure 3. They should be shown together with the atomic model. I strongly recommend removing the bromodomain cartoons. Related with this figure, it would be nice to give more details about the dimensions of this 3DR, scale bars and a brief description of the Cryo-EM structure of Abo1 in the main text.

→ In the revised manuscript, we removed the bromodomain cartoons and we have incorporated the reviewer's suggestions into revised Figure 3 to explain the structure in more detail in the main text.

4. Figure 4: bottom views are missing (indicated in the legend but not shown) and I suggest presenting as well the Apo-Abo1 3DR. Regarding ADP-Abo1 and Apo-Abo1 3DRs, it seems that individual subunits in both states do not show any significant differences in their structures, supporting the idea that both states are symmetric. Have the authors thought to apply six-fold symmetry in order to improve the resolution of the AAA domains and obtain a better definition of the bromodomains?

→ We have included bottom views and electron density maps of apo-Abo1 as suggested in the revised manuscript. We agree that the Apo-Abo1 and ADP-Abo1 seem grossly symmetric, but there are subtle differences in subunits that are perhaps the reason why applying C6 symmetry only worsens the reconstruction. After obtaining a better resolution model of the bromodomain with a Volta phase plate collected dataset, we realized that the pseudo-symmetry axes of the AAA domains and the bromodomains also do not align with each other, and might also contribute to worse results with applied symmetry.

5. Figure 5: providing the angle measurements and the hinges used to show the movement of the subunits would allow a better understanding of the transition.

→ We have included these measurements in figure and legend 5C and the text on pg.18-19.

6. Regarding the real-time ATP-dependent structural changes of Abo1 studied by HS-AFM I wonder if the subunit activation is really random. It seems that subunit 5 in Figure 6e stays open much longer than the rest of subunits. I observed similar behavior in the examples of supplementary Figure 13. Have the authors analyzed the possibility that one of the subunits is

actually favored to initiate activation?

→ In the HS-AFM assays, we do observe that the certain subunits are preferentially activated compared to other subunits, and that the distribution of activation is not completely equal among subunits. This might in part be due to how the Abo1 molecule is attached to the surface. Despite the preference of some subunits to activate more than others, this does not contradict our observations that activation occurs mostly one subunit at a time without a defined order.

7. Related with the substrate binding, I would suggest another representation for the substrate in order to identify clearly the density designated to the substrate. No clear extra mass can be observed in figure 7a as it is. Besides, without any confirmation by mass spectrometry the statement “the extra density in the central pore corresponded to a part of endogenous histone H3-H4 from insect cells” needs to be toned down. Wasn’t there any protein at all detected by MALDI-MS in addition to Abo1 that might account for that mass?

→ As per the reviewer’s suggestions, we have redrawn Figure 7a and b to highlight the extra density in the revised manuscript. As mentioned in the manuscript we did perform simple mass ID experiments of purified Abo1 samples in attempt to identify potentially interesting substrates that might correspond to the extra density. We include a list of the top 30 identified proteins based on the number of unique identified peptides below, but as one can see, the hits include a generic list of highly expressed proteins such as tubulin and translation factors, and do not lend huge insight into the identity of the extra density in our structure.

Based on functional evidence that Abo1 is a histone chaperone, mutational studies showing that substrate binding residues (W345 & E345A) disrupt histone deposition onto DNA, and XL-MS data of Abo1-H3H4, we initially concluded that the extra density is the histone tail. However, we admit that the possibility still exists that the substrate within the Abo1 pore is a protein other than histone, and we have toned down our original statement to: “we attributed the extra density to the binding of an unknown endogenous substrate from insect cells that had been inadvertently co-purified with Walker B mutant Abo1.”, and “...we suspected that some of the extra density in the central pore might correspond to histones or other basic proteins”

No.	Peptide number	ID	Protein name
1	29	TNI009195-RA	Ubiquitin carboxyl-terminal hydrolase puf
2	27	TNI003809-RA	E3 ubiquitin-protein ligase UBR5 isoform X12
3	26	TNI013383-RB	Eukaryotic translation initiation factor 3 subunit A isoform X1
4	26	TNI012432-RB	Eukaryotic translation initiation factor 3 subunit E
5	25	TNI013263-RA	Eukaryotic translation initiation factor 3 subunit C
6	23	TNI005089-RA	Tubulin beta chain
7	17	TNI000739-RA	Eukaryotic translation initiation factor 3 subunit L
8	17	TNI003533-RC	ATP-dependent helicase brm
9	16	TNI008676-RA	Tubulin alpha chain
10	16	TNI003063-RA	Eukaryotic translation initiation factor 3 subunit D
11	16	TNI007071-RC	Ankyrin repeat and KH domain-containing protein 1 isoform X15
12	12	TNI001821-RA	Eukaryotic translation initiation factor 3 subunit M
13	12	TNI009865-RA	SWI/SNF complex subunit SMARCC2
14	11	TNI002442-RB	Tubulin beta chain-like
15	11	TNI005800-RA	Eukaryotic translation initiation factor 3 subunit B
16	11	TNI016717-RA	Heat shock 70 kDa protein cognate 4

17	10	TNI002235-RA	Brahma-associated protein of 60 kDa isoform X3
18	10	TNI012961-RB	Neurobeachin isoform X7
19	9	TNI005125-RB	Eukaryotic translation initiation factor 3 subunit I
20	9	TNI011543-RA	Eukaryotic translation initiation factor 3 subunit H
21	9	TNI010655-RH	SWI/SNF-related matrix-associated actin-dependent regulator of chromatin subfamily E member 1-like isoform X1
22	9	TNI008094-RA	Hrp65 protein-like
23	9	TNI014097-RD	Paired amphipathic helix protein Sin3a-like
24	9	TNI012957-RA	Uncharacterized protein LOC113499142
25	9	TNI000542-RA	cAMP-dependent protein kinase type II regulatory subunit
26	8	TNI009809-RB	Trithorax group protein osa
27	8	TNI006504-RA	Actin, muscle
28	8	TNI002092-RA	Tudor domain-containing protein 3-like isoform X1
29	7	TNI013371-RA	Eukaryotic translation initiation factor 4E transporter-like isoform X3
30	7	TNI008633-RA	rho GTPase-activating protein 190 isoform X1

8. More information about H3-H4 histones used in the cross-linking experiments would be required either in the main text or methods sections. Why did the authors purify the histones from *Xenopus*? Have they attempted purification from yeast?

→ Histones are highly conserved among species such that yeast histones are highly similar sequence *Xenopus* histones (For H3, the sequence identity is 89% and sequence similarity is 96%, while for H4, the sequence identity is 91% and sequence similarity is 98%). For historical reasons, *Xenopus* histones have been the histones of choice in the chromatin field, and continue to be widely used in most chromatin biochemistry and structural studies, such as in the structure of the histone chaperone Asf-1 (English et al (2006)) or in mechanistic studies of the CAF-1 histone chaperone (Mattioli et al (2017)). We included this information in the methods section.

9. Having such clear crosslinks between H3 and AAA1, AAA2 and the bromodomains and, to a lesser extent, H4 and the bromodomains, I would like to know why the reconstruction of this complex has not been attempted.

→ As we have mentioned above, we would love to obtain a cryo-EM structure of Abo1 in complex with histones. We have attempted this approach, but have not been able to obtain an Abo1-H3H4 structure to date.

Specific comments related to Cryo-EM data:

1. In my opinion, the strongest results from the paper are those related to EM. For that reason, I recommend providing more details regarding image processing and 3D reconstruction. Information concerning the selection of 2D and 3D classes or the generation of initial models is missing and would help to evaluate the quality of the processing.

→ We have included more details on the processing of EM data in the methods.

2. Again, I am concerned that image processing has not been optimized to its maximum potential. Some 2D averages in Apo and ADP states should allow a much better definition of the bromodomains. The use of tools from Relion package like Multi-body, Subtraction or the application of masks can help to define the most flexible regions. Besides, sharpening tools like

LocalScale and LocalDeblur can help in the interpretations of 3D reconstructions with areas of different resolution.

→ We have tried to improve the cryo-EM structures by all these methods in Relion, including multi-body refinement, subtraction, and masking. Unfortunately, none of these methods gave better models. Instead, we performed extensive processing (further 2D/ 3D classification) with another Abo1 dataset in the apo state that was collected with a Volta phase plate and we were able to obtain a model with better density for the bromodomain. We include this model in Figure 4a and Supplementary Figure 11 of the revised manuscript, and describe the overall geometry of the bromodomains with respect to the AAA+ domains.

3. Also mentioned before, the application of 6-fold symmetry to the Apo and ADP reconstructions should allow obtaining a better definition of the different domains.

→ We tried applying C6 symmetry to our apo- and ADP-structures, but this only resulted in worse 3D reconstructions, which is due to the fact that these structures are not truly symmetric, and have subtle differences in the subunit structure. When performing extensive processing with another apo-Abo1 dataset, we found that these pseudo-symmetry axis of the bromodomains and AAA+ domains do not align well and likely affect symmetry operations.

Specific comments related to Model Building:

1. In the validation reports for the different structures, it seems that the near full-length mutant reported in the main text was used only for the ATP state (837 residues), whereas a sequence with 1190 residues, that must account for the full-length protein, has been used for Apo and ADP states. Can the authors explain the reason for the use of the different versions of the protein?

→ We apologize for the confusion. This is simply a glitch that arose from the PDB deposition/validation reporting process. The full sequence of the Abo1 is 1190 residues, and for some reason, the report shows that number for the apo and ADP states. But in reality, all 3 models (ATP, ADP, and apo states) contain a build of 837 residues, as the models for the ADP and apo states were derived from the ATP state by MDFF. We have amended the PDB depositions to more accurately reflect these facts.

2. I believe that the model building can be improved using manual adjustment and real-space refinement with Coot to increase the quality of the fitting. This way outliers and angle deviation values should be reduced to meet the standards required for this kind of structures. This is strongly recommended for Apo and ADP structures. I am concerned about the low correlation coefficient (0.6) between the map and the model of the Apo state. These parameters should be improved.

→ We were initially puzzled by the low correlation coefficient that was calculated by Phenix as the observed fit of the MDFF models to the electron density maps was quite good (as shown in Supp Fig. 13), and was similar between the apo and ADP states. By looking closer into the models, we tried removing extra hydrogen atoms introduced by MDFF, and removing extra electron density corresponding to the bromodomain, and found that the correlation coefficient was increased to 0.80 (ADP) and 0.78 (apo).

With regards to manual adjustment and real-space refinement, we believe that at the current resolution of the apo- and ADP- maps, it is more appropriate to use MDFF that provides a non-biased fitting of the model rather than to manually adjust and bias the model by human intervention.

Reviewer #3 (Remarks to the Author):

The manuscript by Cho, Jang et al examines loading of histone H3/H4 dimers by the AAA+ ATPase nucleosome chaperone Abo1 in *S. pombe* by a variety of structural and single molecule methods. Single particle EM reconstructions of several nucleotide bound states of the hexameric Abo1 complex have been generated at 3.5-4.5Å. This seems to be the first structure of an AAA+ atpase from the class that also contains a bromodomain and is therefore thought to be a nucleosome assembly/dissassembly chaperone. Although the bromodomain remains unmodeled here, the atomistic model of the ATPase domains show features that are distinct from other AAA+ atpases and suggest a model of binding the histone tails in the central pore of the ring.

Generally, this paper seems methodologically solid and represents an advance in this field (which admittedly is not my area of expertise). I do have some concerns/questions.

→ We thank the reviewer for his/her comments, many of which relate to the credibility of the CLMS data. We believe that many of these concerns were raised because of our choice to provide only an abbreviated report of the CLMS methods and results. Overall, we have provided more details on our experimental results which we hope convince the reviewer of the significance of our data.

1) Reference 9 reports high homology between Abo1 and Abo2 and they are found to interact via synthetic lethality assay. Rvb1/Rvb2 are AAA+ atpases that in yeast form alternating Rvb1/Rvb2 hexamers. Is the EM reconstruction at sufficient resolution that it conclusively rules out such an arrangement with Abo1/Abo2?

→ We obtained Abo1 protein by heterologous recombinant protein expression from insect cells, and do not expect the presence of any Abo2 protein (a *S. pombe*-specific protein) within the preparation. Also, because we solved the cryo-EM structure to 3.5 Å resolution, we were able to resolve most side chains (Supplementary Fig. 5) and assign individual amino acids of Abo1 with high confidence. Therefore, we are positive that the structure is that of Abo1 alone, and not that of Abo1/Abo2.

2) All of the histone binding assays and structures seem to be done with unmodified histones. The presence of the bromodomain suggests that the endogenous substrates would acetylated somewhere along the tails. The paper would be stronger if it examined binding of Abo1 to acetylated histone substrates.

→ We agree that the effect of histone acetylation on Abo1 would be an interesting avenue to explore considering that bromodomains in general specifically recognize acetylated lysines. However, the specific target of the Abo1 bromodomain has not yet been identified, nor the functional significance of histone acetylations for Abo1 been examined in cells. Moreover, Abo1 has a bromodomain that is non-canonical in several aspects as described at the below.

To explain the non-canonical nature of the ATAD2/Abo1 bromodomain in more detail,

1) There are 6 bromodomains in ATAD2 that are forced into hexameric arrangement (new Supplementary Fig. 12) due to the hexameric ring structure of the AAA+ base of ATAD2. This arrangement would increase the affinity for histones compared to other bromodomain-containing proteins that usually contain only 1 bromodomain per molecule.

2) Judging from our crosslinking-MS data, ATAD2 interacts with histones not only by interactions of the bromodomain with the histone tails, but also by interactions between the bromodomain and histone body, and interactions between the AAA1 domain and histones. This is consistent with the findings of (Koo et al, (2015)) where the authors measured a micromolar

affinity of histone peptides for the isolated ATAD2 bromodomain, but nanomolar affinity of histone peptides for a longer construct of ATAD2 that encompasses the AAA+ and bromodomains. This might also be related to the fact that the ATAD2 bromodomain has a highly negative electrostatic surface potential (Filippakopoulos et al (2012)), and could potentially promote electrostatic interactions between the histone body and ATAD2. Furthermore, this is also consistent with the findings of Gradolatto et al (2009), where the authors found that Yta7 contains histone binding regions outside of the bromodomain.

3) There is still some controversy as to which acetyllysine(s) are the major target(s) of ANCCA. Koo et al (2015) and Morozumi et al (2015) identified H4K5ac and H4K12ac as major targets of the ATAD2 bromodomain, while Revenko et al (2010) found that H3K14ac preferentially binds the ATAD2 bromodomain. In the case of the budding yeast ortholog, Yta7, Gradolatto et al (2009) found that bromodomain binding to histones was unaffected by posttranslational modifications.

4) Adding to this complexity, comparison of yeast Abo1 and human ATAD2 amino acid sequence shows that Abo1 diverges significantly from ATAD2 and other bromodomains. Secondary structure predictions predict that Abo1 adopts the same overall 4-helical bundle fold as other bromodomains, but Abo1 notably diverges in key residues that have been shown to be essential for acetyllysine recognition such as 1021Y and 1064N of ATAD2. In addition, the effects of histone acetylation on Abo1 activity have not been tested, and the target of the Abo1 bromodomain have not been identified.

These observations together establish that ATAD2/Abo1 diverge in character from conventional bromodomain proteins, and thus require full in-depth studies of their characteristics. We plan to pursue future studies relating to the identification of Abo1-specific acetylated histone peptides and the effect of histone acetylation on Abo1 function, but at this point, we believe this is beyond the scope of the current study.

3) The inability to identify the central pore substrate (figure 7) is disappointing. The CLMS results and the analysis of the W345A and E385A mutant doesn't convince me that this histone tails bind directly in the channel. CLMS can be measuring multiple configuration/conformations states of the complex simulataneously and its not clear how the bromodomain and the AAA1 domain can bind the same molecule. The fact that the crosslink between H3-K4 and Abo1-K344 has the highest confidence score does not mean anything except that this particular crosslinked peptide pair ionizes and fragments well. There is only a single crosslink implicating this residue, whereas a number of crosslinks are found between the H3-tail and the C-term side of AAA1 as well as the bromodomain.

→ We agree with the reviewer that the CLMS data could represent the sum of multiple structural conformations. However, we do believe that CLMS data reflect the fact that Abo1 can bind H3H4 by AAA1 and bromodomains (although possibly not simultaneously) for the following reasons. First, we see that the bromodomains line the surface of the AAA1 ring and pack closely with AAA1 domains in the cryo-EM structure (new Figure 4a, apo state), thus suggesting that histones can be nestled in the bromodomain ring adjacent to the AAA+ ring pore. Second, in agreement with this idea, Gradolatto et al (2009) have shown with the budding yeast ortholog, Yta7, that histone binding occurs not only through the bromodomain, but other structural elements outside the bromodomain.

Apart from the CLMS data, we have new data confirming that removal of histone N-terminal tails as well as a mutation in the bromodomain affects Abo1-dependent histone loading onto DNA, similar to Abo1 pore loop mutants (Figure 7F and Supplementary Fig. 19). Thus, we propose that Abo1 pore-histone H3 interactions are likely required for Abo1-H3-H4 loading activity based

on the crosslink between the Abo1 pore and the H3 N-terminus, and the similar defects in Abo1 pore mutants and H3/H4 N-terminal truncated mutants.

The mutation analysis indicates that the mutants bind H3-H4 and hydrolyse ATP at same level as wt while not assembling H3-H4 onto DNA. It could be that some histone tails must bind in the central pore as part of the mechanistic cycle, or it could just be that these residues are necessary for dna loading for some other mechanistic reason.

→ We think that the ATP hydrolysis and histone binding might be coordinated for the mechanistic cycle or DNA binding. However, at this moment, the exact mechanism regarding substrate binding is not clear and requires further extensive biophysical studies. To reflect these mechanistic uncertainties and one of the reviewer's suggestions, we have removed the substrate binding model figure from Figure 7.

4) The CLMS data is not sufficiently described. The sample for CLMS is prepared with quite a high excess of histone (it is not clear if the molar excess of 1:3 is to Abo1 monomer or hexamer in the methods). Also, no FDR is reported for the CLMS dataset and since the search is against a very small sequence database it is prone to false positives as the distribution of decoy crosslink hits cannot be assessed very well. Overall numbers of crosslinks found and distribution between proteins is not reported.

→ The sample for CLMS was prepared with a Abo1 hexamer : histone ratio = 1:3. (All ratios and concentrations reported in the paper are with regards to the hexamer form, as Abo1 exists as a stable hexamer). We chose this ratio as it was the ratio at which most Abo1-histone H3-H4 complexes were formed as judged by native gel analysis. For clarify the quality of the crosslinked sample, we included the native gel as Supplementary Fig. 18.

As we filtered our CLMS data with a relatively stringent criterion, where the xQuest score cutoff (Id-score) was 20. In a previous evaluation of the xQuest program, an Id-score cut off of 20 corresponded to an FDR of less than 3% (Walzthoeni et al, 2012), and an analysis of all of our filtered targets had an FDR of less than 1%. We have added Supplemental Table 3, which summarizes Abo1 intramolecular crosslinks, and a full spreadsheet of all filtered xQuest analysis results for a more complete report of our CLMS data.

5) It is inconceivable that there were no Abo1-Abo1 crosslinks discovered. The fidelity of the CLMS data should be assessed by measuring the distance distribution of the measureable crosslinks against the reported structural model. While MS cannot easily distinguish which pair of protomers produce a given crosslink, it is common practice in this field to measure all 5-possible Lys-Lys distances and reported the most likely (shortest distance). Hence the violation rate of the dataset can be assessed. Additionally, the crosslinking data should provide more information about the localization of the bromodomain.

→ We indeed have many intramolecular Abo1-Abo1 crosslinks, as well as H3-H3 and H4-H4 crosslinks, but chose only to report intermolecular Abo1-H3H4 crosslinks in our initial manuscript as these were the crosslinks that were relevant to our model. We fully agree with the reviewer that there might be interesting structural information buried in the intramolecular Abo1-Abo1 crosslinking data, and in the revised manuscript we provide a list of filtered intramolecular crosslinks (with an Id score >20) in Supplemental Table S3, as well as the full list of identified crosslinks as an Excel file in Supplementary data. However, this data in itself does not provide much useful cross-validation for our cryo-EM model because:

1) Most of the crosslinks are within the bromodomain or between the AAA1 domain and bromodomain. As we were not able to obtain a structure of the bromodomain, nor approximately fit a human ATAD2 bromodomain structure-based homology model into our cryo-EM map, we are unable to cross-validate most crosslinks with our cryo-EM structure.

2) Even for crosslinks that are formed within AAA domains, the interpretation of these crosslinks is extremely difficult because they could be crosslinks formed within a single AAA domain (intra-subunit) or a crosslink formed between one AAA domain and any of 5 other AAA domains (inter-subunit).

With those caveats in mind, when we analyze the crosslinks that are mappable to the cryo-EM maps, 1) there are no crosslinks between the AAA2 domain and bromodomain or the AAA2 domains and histones, where the distances are too far to be crosslinked, and 2) several specific crosslinks exist between the AAA1 domain and bromodomain, where the bromodomain is located near the AAA1 domain based on our structure of Abo1 in the apo state showing the location of the bromodomains. These observations are overall consistent with our cryo-EM structures, and provide confidence that histone H3-H4 is specifically binding on the “top” surface of Abo1.

In short, I believe the paper is fairly high quality but I find the substrate binding model (Fig 7C) not particularly likely. The paper could be improved by further exploration of the substrate characteristics in terms of histone acetylation, stoichiometry of histone binding, and by trying to identify a DNA binding region of the complex. The CLMS data is poorly reported, and seemingly relevant crosslinks are hidden from discussion when they contain other relevant information to the paper.

→ Because we only have a preliminary understanding of how Abo1 binds histones and loads them onto DNA, we have removed the speculative substrate binding model from Figure 7. As the reviewer pointed out, it would be greatly interesting to investigate the substrate characteristics and the substrate binding mode of Abo1. However, we believe that this is beyond the scope of the current manuscript describing the first cryo-EM structures of Abo1, and should be reserved for future work dissecting the structural basis of Abo1 binding and DNA deposition

Reviewer #4 (Remarks to the Author):

In this manuscript, Cho and colleagues report a series of CryoEM structures of the Abo1 AAA+ ATPase. The ATP-saturated structure of the Walker B mutant shows a broken ring/washer hexamer with ATP molecules located at the interfaces between the protomer (except for the broken interface). Resolution of CryoEM map for this complex allowed the authors to build the structure of the ATPase part (sans bromodomain) de novo. The ADP-bound complex was visualized at lower resolution, but sufficient to reveal a more symmetric, closed ring structure. A number of features conserved among AAA+ ATPase and those unique to Abo1 are discussed. These structures will be important to the AAA+ ATPase field and will advance our understanding of the histone chaperones. Most interestingly, the high speed AFM imaging of the wild type Abo1 in the presence of ATP confirmed that the hexamer transitions between the closed ring and the open washer conformations with interface breaking at random subunit, which the authors argue corresponds to the ATP hydrolysis.

> We thank the reviewer for the assessment and are happy to address the issues raised below:

1. These are very impressive data, but it would be very informative if the authors show representative images of the open complexes where more than one subunit is off the mica plain. The time series in Fig 6E and in Supplemental Fig. 13 suggest that the ring opening occasionally occurs simultaneously at multiple subunits, and even at non adjacent subunits (e.g. 2,3,5 in Fig 6E). Without an AFM frame, it is difficult to imagine this event without the hexamer completely dissociating, as the interface in the CryoEM structure seems to be broken across both AAA1 and AAA2 domains.

> We include more images of open complexes where more than one subunit seems to disappear from the AFM field of view in Supplementary Fig. 14. However, we note that compared to the full length of the AFM movie, events where more than one subunit disappears from the field of view are relatively rare and transient. Nonetheless, it is possible that dissociation of adjacent Abo1 subunits can take place so that the Abo1 hexamer is broken. However, because we see recovery to a symmetric hexamer after these events, we believe these conformations must be very transient. The Abo1 molecule where multiple lobes seem to disappear goes through dynamic motions when more than 2 lobes are absent. In some of these frames the hexamer seems to rotate such that we observe side-on views of the AAA ring. In addition, please refer movie files of AFM data included as Supplementary material.

2. Another point here. From the time series of the ring openings observed by AFM, the authors should be able to estimate a k_{cat} for ATP hydrolysis. Does it correspond to the ATPase activity measured in solution? A good correspondence here would straighten the authors' model where the opening of the ring corresponds to the ATP hydrolysis.

> According to the reviewer's suggestion, we analyzed the dwell time distribution of our AFM data, and calculated the rate of ATP hydrolysis from this data. (Data is shown in Supplemental Fig. 15) Based on this analysis, we obtain a k_{cat} of $1.5s^{-1}$ for the open state and $0.99s^{-1}$ for the closed state, which agrees roughly with the rate of $0.8s^{-1}$ measured in solution.

3. In addition to structural information, the authors also report functional studies, as it has been unclear what the Abo1 function is with respect to histone H3-H4. The authors show that Abo1 interacts with H3-H4 and promotes its deposition on DNA in an ATP-dependent manner. The latter was investigated using single-molecule DNA curtain technique. This is an important piece of data in the manuscript and qualitatively, the conclusions seem correct – Abo1 loads, but does not unload H3-H4 onto the DNA and the ATP hydrolysis is indeed important. Only minimal quantification of the single-molecule experiments, however, is presented and a lot of important information is missing in the methods.

> We have analyzed our single-molecule data quantitatively and included statistical data for the DNA curtain assays in the revised version (Supplemental figures 2a and 3), which should help solidify our results.

4. How many molecules were analyzed in each experiment?

> We have included the number of analyzed molecules for all single-molecule experiments in the Figure legend 2 and Supplementary Figure 2.

5. Why particular concentrations of Abo1 and H3-H4 were selected? Considering a K_d of about 23 nM, only a small fraction of Abo1 will be in complex with H3-H4. Is it expected to work catalytically?

> The protein concentration for the DNA curtain assays was determined by finding the maximum concentration at which significant non-specific binding to the surface did not occur. Although DNA curtains are based on a lipid bilayer that blocks non-specific binding, at high protein concentrations, non-specific binding does occur and can be observed by the inability of DNA molecules to recoil toward the barrier when flow is switched off. The ratio of Abo1:H3-H4=1:2.5 was selected because we found that Abo1-H3H4 complex formation was maximal at this ratio as judged by native gel titration and analysis (Note that H3-H4 is in excess of Abo1, as opposed to the opposite situation in the K_d analysis with fluorescent histones.). Thus, we expect that most Abo1 molecules would be in complex with H3-H4.

6. Some non-specific binding of H3-H4 to the surface can be observed (Fig. 2D “Flow off” panel). How was this non-specific binding distinguished from the actual binding to DNA?

> One of the hallmarks of the DNA curtain assay (Fazio et al, 2008) is that it is able to distinguish between non-specific binding of molecules to the surface from specific binding to DNA curtains by switching flow on and off. H3-H4 non-specifically bound to the surface remains in the same position regardless of flow. In contrast, H3-H4 bound to DNA curtains disappears from the field of view when flow is switched off, as DNA coils toward the barrier and is placed outside of the excitation area created by the evanescent field. As shown in Fig. 2C, we analyzed only the H3-H4 molecules that disappeared when flow was switched off.

7. Is there any specificity to the H3-H4 deposition, or is it completely random?

> In the revised manuscript, we have added the binding distribution histogram of H3-H4 dimers on lambda DNA (Supplementary Fig. 2A), which demonstrates that H3-H4 binding is random, and is sequence-independent as expected for general histone chaperones.

8. Minor point. The manuscript can benefit from some editing. Many sentences throughout the manuscript are difficult to understand, which detracts from the beautiful data the authors try to communicate.

> We have tried to revise parts of the manuscript and polish some sentences that might have been difficult to understand.

REVIEWERS' COMMENTS:

Reviewer #1 (Remarks to the Author):

The authors have done an outstanding job at responding to my previous concerns. In particular, the additional MNase studies, bromodomain mutations, and histone tail truncations solidify their original conclusions. I also appreciate that they have better balanced their conclusions regarding tetrasome assembly. This is now a highly significant piece.

Reviewer #2 (Remarks to the Author):

The revision presented by the authors improves the interpretation of the results significantly. All major points have been addressed, or tried to. In this version, the key point that Abo1 clearly interacts with H3-H4 and loads them into the DNA when ATP is present is fully supported.

The 3D reconstruction of the bromodomains has been improved by performing further Cryo-EM using a Volta Phase Plate, in an attempt to increase the resolution. Although unfortunately not as successful as it could, the effort to perform this new reconstruction is really appreciated. Now, 6 lobules are clearly observed which must correspond to these domains, but the limited resolution has prevented the model building in these regions.

I still think that the nucleotide state has an influence in the stabilization of the bromodomains, due to the difference in resolution in the different conformations. I suggest the authors should mention this in the main text.

The part describing the protein that co-purifies with Abo1 has been modulated and now sounds more reasonable

There are a couple of errors, line 357 "the rings were subjected", line 440 change "," for "."

No mention is made to the methods used for the de novo model building

Reviewer #3 (Remarks to the Author):

The revised manuscript includes additional reporting on the XLMS data, as well as new deposition experiments with the "tail-less" histones and Abo1 pore/bromodomain mutants. The authors have also toned down some of their claims regarding identifying density corresponding to a histone-tail in the central cavity as well as some of the other claims of the paper (eg regarding "assembling" vs "deposition" of tetrasomes).

Overall, I think this is a high quality, interesting manuscript that warrants publication. I will note that the XLMS data is still being under-utilized... the problems they mention in the rebuttal letter can be accounted for. On a technical note, FDR analysis using a small decoy database (3 randomized sequences) is not large enough to accurately capture the distribution of random matches. However, the XLMS contributes only a small part of the overall story and the AAA1 domain to histone crosslinks are likely correct, although its hard to assess this without access to the spectra.

Reviewer #4 (Remarks to the Author):

The authors adequately addressed all my previous concerns. The manuscript was very much improved and strengthened by the additional data corroborating the authors conclusions.

RESPONSE TO REVIEWERS' COMMENTS:

Reviewer #1 (Remarks to the Author):

The authors have done an outstanding job at responding to my previous concerns. In particular, the additional MNase studies, bromodomain mutations, and histone tail truncations solidify their original conclusions. I also appreciate that they have better balanced their conclusions regarding tetrasome assembly. This is now a highly significant piece.

→ We appreciate the constructive criticisms raised by the reviewer and agree that the suggested experiments have improved the revised manuscript.

Reviewer #2 (Remarks to the Author):

The revision presented by the authors improves the interpretation of the results significantly. All major points have been addressed, or tried to. In this version, the key point that Abo1 clearly interacts with H3-H4 and loads them into the DNA when ATP is present is fully supported.

The 3D reconstruction of the bromodomains has been improved by performing further Cryo-EM using a Volta Phase Plate, in an attempt to increase the resolution. Although unfortunately not as successful as it could, the effort to perform this new reconstruction is really appreciated. Now, 6 lobules are clearly observed which must correspond to these domains, but the limited resolution has prevented the model building in these regions.

I still think that the nucleotide state has an influence in the stabilization of the bromodomains, due to the difference in resolution in the different conformations. I suggest the authors should mention this in the main text.

→ We added a sentence in the discussion: “We speculate that such nucleotide-dependent changes in AAA+ ring structure regulate the structure of the bromodomain ring, as the bromodomain density in the apo and ADP states is more distinct compared to the ATP state.”

The part describing the protein that co-purifies with Abo1 has been modulated and now sounds more reasonable.

→ We are glad that the reviewer’s concerns have been resolved.

There are a couple of errors, line 357 “the rings were subjected”, line 440 change “,” for “.”

→ We have corrected the mentioned errors.

No mention is made to the methods used for the de novo model building

→ We have added a section in the Methods titled “atomic model building and refinement”.

Reviewer #3 (Remarks to the Author):

The revised manuscript includes additional reporting on the XLMS data, as well as new deposition experiments with the "tail-less" histones and Abo1 pore/bromodomain mutants. The authors have also toned down some of their claims regarding identifying density corresponding to a histone-tail in the central cavity as well as some of the other claims of the paper (eg regarding "assembling" vs "deposition" of tetrasomes).

Overall, I think this is a high quality, interesting manuscript that warrants publication. I will note that the XLMS data is still being under-utilized... the problems they mention in the rebuttal letter can be accounted for. On a technical note, FDR analysis using a small decoy database (3 randomized sequences) is not large enough to accurately capture the distribution of random matches. However, the XLMS contributes only a small part of the overall story and the AAA1 domain to histone crosslinks are likely correct, although its hard to assess this without access to the spectra.

→ We appreciate the reviewer's comments. We agree that there is likely more information to be mined in the XLMS dataset but we believe that such information should be validated with other experimental methods, which is better reserved for future studies.

Reviewer #4 (Remarks to the Author):

The authors adequately addressed all my previous concerns. The manuscript was very much improved and strengthened by the additional data corroborating the authors conclusions.

→ We thank the reviewer for his/her comments.